# An intravenous pancreatic cancer therapeutic: Characterization of CRISPR/Cas9n-modified *Clostridium novyi*-Non Toxic

Kaitlin M. Dailey[1,2,3]*, James M. Small[4], Jessica E. Pullan[3,5], Seth Winfree[1,6], Krysten E. Vance[1], Megan Orr[7,8], Sanku Mallik[2,3], Kenneth W. Bayles[6], Michael A. Hollingsworth[1], Amanda E. Brooks[2,3,9]

1 Eppley Institute for Research in Cancer, University of Nebraska Medical Center, Omaha, NE, United States of America, 2 Cell and Molecular Biology Program, North Dakota State University, Fargo, ND, United States of America, 3 Department of Pharmaceutical Sciences, North Dakota State University, Fargo, ND, United States of America, 4 Department of Pathology and Microbiology, Rocky Vista University, Parker, CO, United States of America, 5 Department of Physical Science, Southern Utah University, Cedar City, UT, United States of America, 6 Department of Pathology and Microbiology, University of Nebraska Medical Center, Omaha, NE, United States of America, 7 Department of Statistics, North Dakota State University, Fargo, ND, United States of America, 8 Center for Diagnostics and Therapeutic Strategies in Pancreatic Cancer Biostatistics Core Facility, North Dakota State University, Fargo, ND, United States of America, 9 Department of Research and Scholarly Activity, Rocky Vista University, Ivins, UT, United States of America

* kadailey@unmc.edu

**Data Availability Statement:** The histology datasets generated during and/or analyzed during the current study are available in the repository: The Cancer Imaging Archive hosted by the National

## Abstract

*Clostridium novyi* has demonstrated selective efficacy against solid tumors largely due to the microenvironment contained within dense tumor cores. The core of a solid tumor is typically hypoxic, acidic, and necrotic—impeding the penetration of current therapeutics. *C. novyi* is attracted to the tumor microenvironment and once there, can both lyse and proliferate while simultaneously re-activating the suppressed immune system. *C. novyi* systemic toxicity is easily mitigated by knocking out the phage DNA plasmid encoded alpha toxin resulting in *C. novyi*-NT; but, after intravenous injection spores are quickly cleared by phagocytosis before accomplishing significant tumor localization. *C. novyi*-NT could be designed to accomplish intravenous delivery with the potential to target all solid tumors and their metastases in a single dose. This study characterizes CRISPR/Cas9 modified *C. novyi*-NT to insert the gene for RGD, a tumor targeting peptide, expressed within the promoter region of a spore coat protein. Expression of the RGD peptide on the outer spore coat of *C. novyi*-NT indicates an increased capacity for tumor localization of *C. novyi* upon intravenous introduction based on the natural binding of RGD with the $\alpha_v\beta_3$ integrin commonly overexpressed on the epithelial tissue surrounding a tumor, and lead to immune stimulation.

## Introduction

After decades of research, the 5-year relative survival rates for most major forms of cancer (*e.g.* breast, colon, melanoma, etc.) have seen significant improvements with an average 85–90% five year survival rate [1, 2]. Despite experiencing the same scientific advancements, pancreatic

Cancer Institute, https://www.cancerimagingarchive.net/. Further datasets generated during and/or analyzed during the current study are included in the supporting information.

**Funding:** We acknowledge funding support for the research within this publication from the Center for Diagnostic and Therapeutic Strategies in Pancreatic Cancer at North Dakota State University 1P20GM109024 to SM and AEB, as well as discretionary funds from MAH and KWB at UNMC. Further funding was provided by a Nebraska Department of Health and Human Services LB506 to MAH, KWB, and KMD. We also gratefully acknowledge the Geographical Management of Cancer Health Disparities Program, Region 3 hosted by the University of New Mexico Comprehensive Cancer Center for covering costs associated with publication. The funders had no role in study design, data collection and analysis, decision to publish, or preparation of the manuscript.

**Competing interests:** The authors have declared that no competing interests exist.

cancer continues to have a relative survival rate of a mere 10% [3] with only a one percent gain in patient's five year survival rate observed for the first time in 2020 since 1980 [4]. Six patients are projected to die from pancreatic cancer within the next hour, with a total of over 48,000 deaths occurring in the United States throughout 2021 [4]. This is particularly concerning since the occurrence of pancreatic cancer is predicted to grow by over 30% in the next twenty years [4]. Further, in the mere six months a pancreatic cancer patient is likely to survive past diagnosis, medical bills of more than $13,000 for each of those months is accrued for a total cost often topping $200,000 after insurance [5]. While substantial studies characterizing tumors have led to impressive novel therapeutics and understanding of tumor development, these glaring statistics and lack of clinical progress beg for the exploration of alternative therapeutics for pancreatic adenocarcinoma. Oncolytic bacteria represent one of the most promising areas of discovery for novel solid tumor therapeutics [6–8], particularly with the advent of 21st century genetic engineering technology.

Oncolytic bacteria are bacteria that are particularly attracted to the characteristics of solid tumors that commonly confound traditional drug delivery efforts: poor vascularity, high acidity, hypoxia and dense desmoplasia [9–12]. Select oncolytic bacterial species have a natural capacity to detect and navigate subtle cytokine, pH, and oxygen gradients with a demonstrated ability to localize specifically to solid tumors from the bloodstream using this intrinsic, ultra-sensitive chemotaxis [6, 7, 13, 14]. Once localized to the tumor center, these bacteria then colonize and not only lyse the tumor from its core, in stark contrast to modern therapeutics, but can also simultaneously activate the suppressed immune system at the tumor margins [15, 16] resulting in a re-training of the immune system and giving rise to potent life-long tumor surveillance [15]. Oncolytic bacteria accomplish this localization through an active, motile mechanism [6, 13] and are therefore not limited by the enhanced permeability and retention (EPR) effect. Furthermore, the initial therapeutic dose does not limit the tumor lytic capacity as proliferation can occur in the harsh tumor core environment, resulting in unique and complex pharmacokinetics.

Among oncolytic bacteria, *Clostridium novyi* is distinct in part because of its strict, obligate micro-anaerobicity and sporulation capacity [17, 18]. *C. novyi* represents a particularly promising oncolytic bacterial species due to its ability to form spores with little to no immunogenicity but with an inherent chemotactic ability [16, 19]. The lifecycle of *C. novyi* consists of two distinct phases: the active, lytic and proliferative vegetative phase, and the largely dormant but still motile spore phase [17, 20]. Importantly, the vegetative cells of *C. novyi* cannot survive in any physiologically relevant level of oxygen [15, 16, 19–21] while the spores are unperturbed in oxygenated environments. Though the spores will survive in the blood stream or healthy tissues, germination to vegetative cells cannot occur until an adequately hypoxic environment is achieved [15, 16, 19]. The culmination of these characteristics leads to the potential to modify *C. novyi* spores for selective, tumor specific therapeutic delivery through systemic administration. These efforts have been furthered by the ease with which *C. novyi* systemic toxicity can be attenuated. While known to cause tissue necrosis, recognized as gas gangrene clinically, removal of the phage DNA plasmid encoding α-toxin fully mitigates sepsis, creating a nontoxic strain known as *C. novyi*-NT [17]. This attenuated form of *C. novyi* has not demonstrated significant decreases in any of the other oncolytic characteristics that make *C. novyi* such a promising oncotherapeutic [17, 18, 21].

*C. novyi*-NT oncotherapy targets physiologic characteristics (*e.g.*, hypoxia, acidity, necrosis) that are shared by nearly every solid tumor as well as small malignant islands such as those that occur during metastasis [11, 12]. Therefore, it is reasonable to predict that this therapy would have efficacy regardless of solid tumor tissue origin, location, or stage. Unfortunately, early studies probing *C. novyi*-NT toxicity found that despite low to negligible dose-related toxicity

after IV administration in mice, spore clearance occurs naturally within 24hrs through phagocytosis without any canonical indications of inflammation [19]. The majority of the initial dose was cleared within one hour–significantly inhibiting tumor localization [19]. Despite this low delivery of approximately 1% of the initial dose, the vast majority of mice were found to have significant tumor mitigation [19]. Thus, current efforts at clinical translation have focused on local administration by directly injecting tumors with spores.

Intratumoral injections (IT) of *C. novyi*-NT spores have demonstrated great promise in pre-clinical studies and recently in a phase II clinical trial as a therapy for refractory solid tumors [22–24]. However, tumors can be small, multiple, or unsafe to access. Systemic intravenous delivery (IV) of *C. novyi*-NT could be developed to overcome this limitation. Theoretically, IV-delivered spores could allow for localization to not only the primary tumor but also to any metastases in a single treatment due to natural selectivity for the unique physiological microenvironment of solid tumor loci. However, for oncolytic efficacy, the bacteria are in a race to localize to the tumor before host immune clearance occurs: a challenging task with low efficiency. In this study, CRISPR/Cas9 was used to modify the genome of *C. novyi* to elicit the insertion, expression, and spore coat insertion of the tumor targeting tri-peptide, Arg-Gly-Asp (RGD). In previous studies in the field of nanoparticle drug delivery, tumor accumulation was improved through the integration of the RGD motif [25, 26]. RGD has a well-characterized affinity for the $\alpha_v\beta_3$ integrin commonly overexpressed on tumor cells and tumor associated epithelium [26, 27] including pancreatic tumors [28]. Further, transient exogenous plasmid expression of RGD on the surface of other oncolytic bacterial species has altered biodistribution advantageously [7, 13, 29]. We have previously accomplished genomic editing to insert the sequence of known tumor targeting tag 'RGD' into *C. novyi*-NT and confirmed retained oncolytic capacities [20]. This manuscript describes the incorporation of RGD into the spore coat, including probing biodistribution and immune stimulation of modified *C. novyi*-NT in a murine pancreatic tumor model with an intact immune system.

## Results

### RGD peptide encoding gene genomic insertion confirmation

Transformation of calcium competent *C. novyi* with the CRISPR plasmid pKMD002 (see Supporting Information) resulted in the growth of more than thirty colonies on selective media plates. Five candidates were selected for genomic DNA isolation and subsequent 16S rRNA PCR amplification. The resulting PCR amplicons were digested with *EcoR*V to establish the successful insertion of the RGD-peptide encoding gene into the *C. novyi* genome. All five candidates demonstrated the presence of two bands after restriction digestion, indicating that all were positive for genomic insertion (see Supporting Information) [20].

### Characterization of RGD-modified *C. novyi* spores *In Vitro*

**Adhesion assay.** To demonstrate the protein expression, physical availability, and functionality of the genetically inserted RGD gene, an adhesion assay was created based on the known RGD integrin $\alpha_V\beta_3$ binding interaction as has been used to characterize nanoparticles [30]. An $\alpha_V\beta_3$ coated surface was generated (Supporting Information) and inoculated with RGD-modified *C. novyi* spores (Fig 1A). Excess spores were washed away, and remaining spores were stained with crystal violet. The excess crystal violet was quantified via absorbance at 590nm, with a lack of signal, indirectly indicating the quantity of spores that had remained dyed by CV on the integrin coated surface (Fig 1B). Candidate A demonstrated a 2-fold change significantly greater than that of both un-modified *C. novyi* spores (used as a control) as well as the other candidates assessed. In a separate but corroborating assay, to directly quantify the

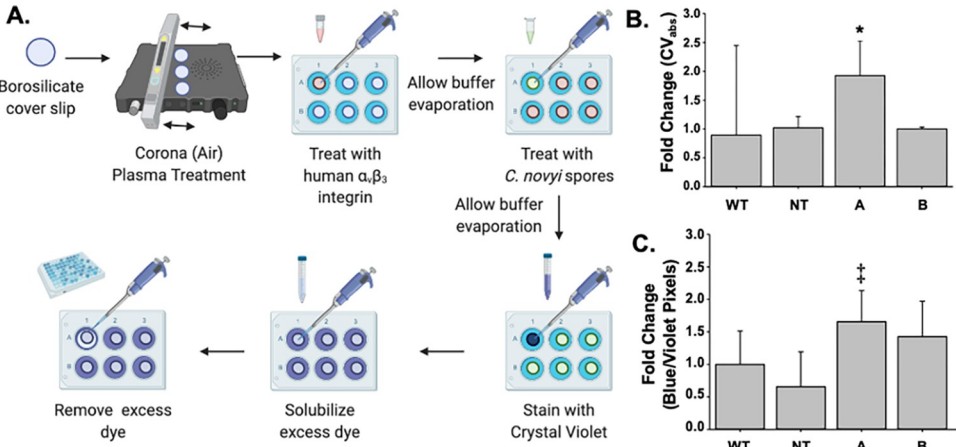

**Fig 1. Establishing functionality of RGD peptide insertion.** (A) Pictural description of the $\alpha_V\beta_3$ adhesion assay methodology. (B) Fold change observed in crystal violet (CV) absorbance at a wavelength of 590 for wild type (WT), non-toxic (NT) *C. novyi* as well as the putative RGD-modified candidates (A and B) after exposure to the $\alpha_V\beta_3$ coated surface of the adhesion assay. * denotes a p value of < 0.05 when compared to any other cohort, including WT, NT, and modification Candidate B. (C) Average CV pixel count of entire integrin coated surface for candidates A and B as well as wild-type (WT) and non-toxic (NT) *C. novyi* that remain on the $\alpha_V\beta_3$ coated surface. ‡ denotes a p value of 0.05 when compared to any other cohort, including WT, NT, and modification Candidate B. Error bars represent standard deviation from the cumulative mean of three experimental replications (n = 6 each) for a total n = 18.

spores remaining on the integrin coated surface, 40X scanning brightfield confocal microscopy images of the entire integrin-coated surface were obtained after crystal violet staining. The blue/violet pixels contained within these images were then counted via FIJI (Fiji Is Just ImageJ, 64-bit 2020), the open source image processing software [31] (representative images are included in Supporting Information). Three experimental replications were conducted with a n = 6 per treatment (RGD-modification candidates, non-modified spores, PBS) for a total n = 18 reported in Fig 1. The resulting quantification from this method corroborates the data generated from measuring the excess CV run off and confirms that Candidate A demonstrates a higher rate of adhesion by the presence of more spores remaining on the surface when compared to any other cohort, including WT, NT, and modification Candidate B (Fig 1B and 1C).

**Transmission electron microscopy (TEM).**    To further probe the expression and availability of the RGD peptide, spores were isolated and purified for transmission electron microscopy (TEM). Typically, spores generated from *C. novyi* vegetative cells are surrounded by a complex spore coat consisting of multiple layers. The outermost layer of the spore consists of a sacculus, then an amorphous shell with intertwined honeycomb layers composed of an amorphous region interleaved with parasporal layers [32]. Frequently, this 'honeycomb' layer has an attachment to the spore coat, which consists of 3–6 layers: undercoat, cortex, germ cell wall, and the spore core. Several landmarks associated with normal spore coat architecture, which had been previously published [32] (Fig 2A, figure modified from [32]), were observed in images of modified spores (Supporting Information). Some layers of the spore architecture were notably absent through this method of preparation; however, others remained. The dark, dense core (Co) and surrounding cortex (Cx) layers were observable, as was the gray layer just outside the cortex indicating the characteristic 5–7 layer coat (Ct) containing the dark staining granular (G) paracrystaline layer. The germ cell wall that distinguished the spore core (Co) from the cortex (Cx) cannot be observed because ultrathin sections were not obtained for this study. Furthermore, the method of isolation utilized in this study, which would be the same as that used for clinical spore preparation, does not result in an intact amorphous layer. When

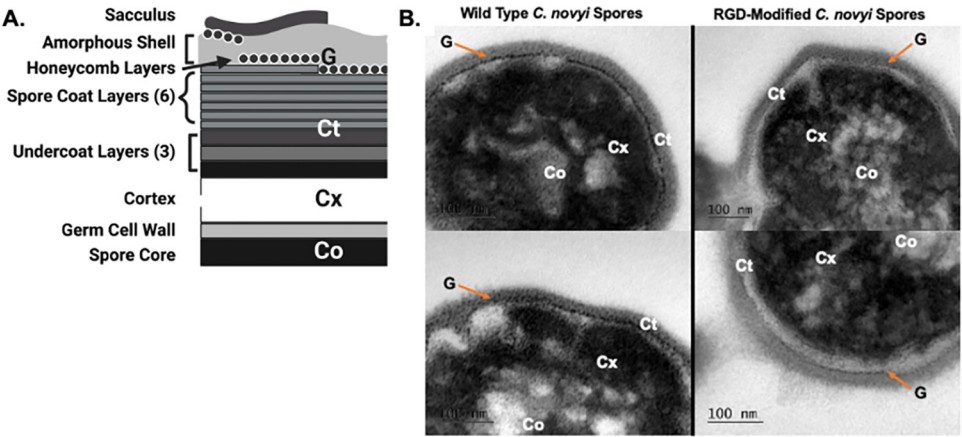

**Fig 2. Physical characterization of the inserted RGD peptide in spore coat.** (A) Schematic representation of the *C. novyi* spore coat layers as previously published [32]. Figure modified from [32]. Note that the method of isolation in this study removes the sacculus. (B) Representative images of transmission electron microscopy images of wild type and RGD-modified candidate A *C. novyi* spores. Sacculus is removed through TEM fixation procedures. All resulting images contained in Supporting Information (Co–core, Cx- cortex, Ct–coat, G–granular layer).

compared to previously published TEM images for unmodified *C. novyi* spores [32] (Fig 2A, figure modified from [32]), the RGD-modified spores showed an observable, disrupted para-crystalline layer as indicated by G on the images (Fig 2B). However, it should be noted that due to the complexity of the many layers of the spore coat, which exact layer contained the RGD peptide cannot be indicated by the methods of this study.

## *In Vivo* model of intravenously injected *C. novyi* spores

**Characterization of RGD-modified *C. novyi* spores *In Vivo*.** Once the genetic and physical insertion of RGD had been confirmed, the pathophysiological effect of this modification was characterized in an immunocompetent C56Bl/6 murine model (Fig 3A). Half of the mice

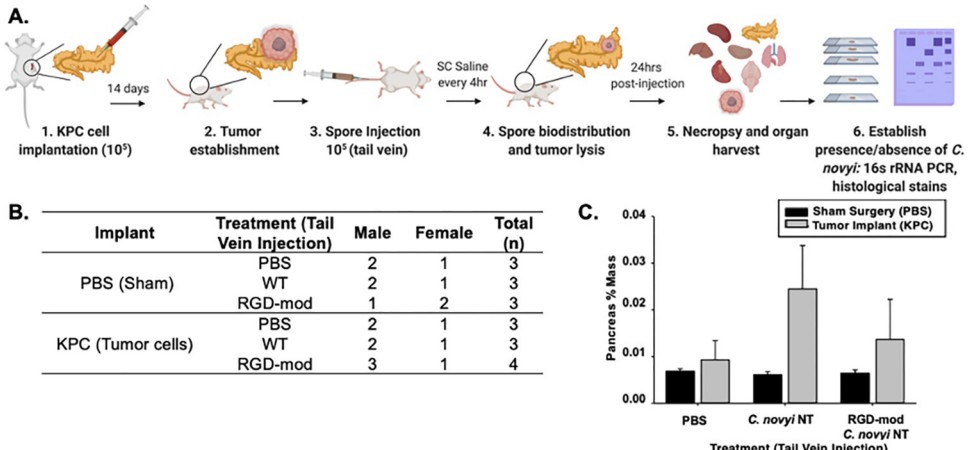

**Fig 3. Characterization of RGD-modified *C. novyi*-NT in murine models.** (A) Surgical procedure utilized to produce the orthotopic pancreatic tumor model in C57BL/6 immunocompetent mice, (B) Representative table of the number and sex of mice in each cohort, (C) The average percent weight of tumors harvested from murine cohorts in B 24hrs after tail vein injections. Note that mass includes inflammation, with swelling of the tumor often being the foremost indication of successful localization. Error bars represent standard deviation from the mean.

(n = 10) underwent a pancreatic tumor implantation surgery with KPC 5508 cells, while the other half (n = 9) underwent a sham surgery that injected PBS instead of tumorigenic cells (Fig 3B). The tenth tumor implanted mouse was initially performed in case of issues during tumor development that would lead to exclusion prior to tail vein treatment. When no such exclusion was necessary, it was randomly assigned to a treatment group, which ended up being the RGD-modified *C. novyi*-NT cohort. Tumors developed over two weeks, then, as an IV injection analog, tail vein injections of *C. novyi*-NT spores, RGD-modified *C. novyi*-NT spores, or PBS were given to each mouse (n = minimum 3 each cohort) (Fig 3A). Mice were observed constantly for 24hrs post exposure to treatments, with temperatures checked every 4hrs when subcutaneous fluids were administered. No blood clots, fever or other signs of distress including sepsis were observed, suggesting no adverse events occurred at any point in this study.

**Pancreas percent mass.** The weights of murine pancreata were observed after sacrifice to probe the efficacy of RGD-modified *C. novyi*-NT to lead to tumor necrosis. The size of a pancreas is influenced by many factors, most prominently the overall weight and sex of the mouse. Therefore, we took the pancreata and associated tumor weight as a percentage of the overall mouse weight to draw comparisons more accurately between cohorts. Surprisingly, the tumor-containing pancreata from mice injected with spores demonstrated a higher percent mass than the pancreas and associated tumor tissue of the cohort that received PBS treatment (Fig 3C), which is often a sign of inflammation indicating successful spore localization [15, 19]. Furthermore, the pancreas percent mass for the cohort injected with RGD-modified spores was observed to be less than that of the non-modified spore injected cohort (Unmodified Spores > RGD-modified spores > PBS).

**Biodistribution of *C. novyi*-NT spores.** Homogenized tissues harvested post-injection underwent genomic isolation and subsequent PCR with primers designed to be specific to the 16S rRNA of *C. novyi*. For the cohort of mice that underwent sham surgeries and PBS tail vein injection, only one amplicon was observed in the lung of a single mouse—likely due to cross contamination in the euthanasia chamber as neither of the other two mice in this cohort demonstrated any presence of *C. novyi* in any tissue (Fig 4A, Supporting Information). When wild type *C. novyi*-NT spores were injected into control mice with sham surgeries, amplicons were observed in the liver and kidney of one mouse, and the spleen and pancreas for another from the three mice within this cohort. The third mouse on this cohort did not have any amplicons present after PCR was conducted. Analogously, 24hrs after injection, RGD-modified *C. novyi*-NT spores were detected in the spleen, pancreas, kidney, lung, heart and brain of two control mice with sham surgeries. In mice implanted with tumors and subsequently injected with PBS, *C. novyi*-NT was not detected in any mouse within the cohort. The presence of unmodified *C. novyi*-NT spores was indicated in the spleen, pancreas and associated tumor tissue, kidney, and lung of two of the three mice in the cohort after tail vein injection. Similarly, in tumor containing mice that received RGD-modified *C. novyi*-NT spores, 16S rRNA primers amplified corresponding DNA in two of four harvested spleens, all four pancreases and associated tumor tissue, in half of the harvested lungs, three quarters of harvested hearts and half of the harvested brains (Fig 4A, Supporting Information).

**Bacterial burden quantification.** To establish the bacterial burden of organs, the genomic DNA isolates underwent normalization and further 16S rRNA PCR analysis for quantification (Supporting Information). The total bacterial burden for each mouse was quantified (Fig 4B), with the nanograms of amplified DNA being notably higher for mice inoculated with RGD-modified spores than those injected with unmodified spores. When the nanograms per amplicon were quantified per organ, the bacterial burden was found to be higher for mice exposed to RGD-modified *C. novyi*-NT spores in almost every organ except the lungs of tumor

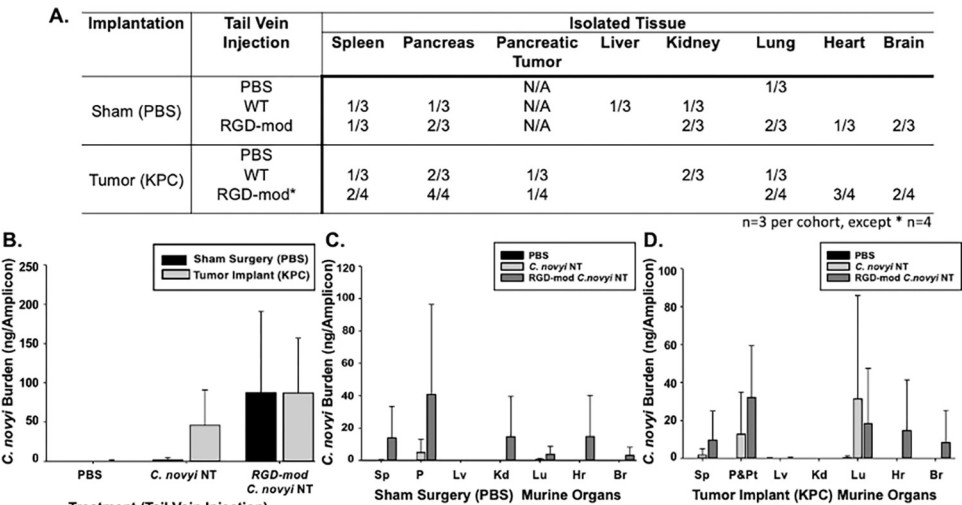

**Fig 4. Quantifying *C. novyi* spores after IV injection.** After tissue harvest and homogenization, bacterial burden was assessed for tissues positive for *C. novyi* specific 16S rRNA. Cohorts employed in this study are in Fig 3B. (A) Biodistribution of *C. novyi* Spores after IV injection. Primers designed specifically for 16S rRNA of *C. novyi* were used to conduct PCR to determine the biodistribution of spores 24hrs after injection. Each number in the table indicates the number of animals per cohort the presence *C. novyi*-NT was detected. (B) Further 16S rRNA PCR was conducted with the template concentration normalized across tissues. The combined bioburden of all major organs within each cohort (WT–wild type *C. novyi* spore tail vein injection, RGD-mod—RGD-modified *C. novyi* tail vein injection) and each major organ within a cohort when (C) sham tumor implantation surgery occurred versus when (D) tumor implantation with KPC cells was conducted. (Sp–spleen, P–pancreas, P&Pt–pancreas and pancreatic tumor, Lv- liver, Kd- kidney, Lu–lung, Hr–heart, Br–brain.) Error bars represent standard deviation from the mean.

containing mice (Fig 4C and 4D). Importantly, a larger quantity of *C. novyi* DNA was detected in the pancreas and tumor associated tissue of the tumor implant cohort that received RGD-modified *C. novyi*-NT spore injections (Fig 4D).

**Histological analysis.** Tissue Gram-stains were initially performed on all sections, but the value of examining Gram-stained slides in the absence of pathologic changes of infection is essentially zero. For that reason, higher resolution scans of gram stains (100X oil emersion) were performed only on areas of tumor with necrosis or acute inflammation. The initial scans of H&E-stained slides at 20X were more than adequate for routine histologic examination, but did not have adequate resolution for finding microorganisms.

A Board-certified pathologist made initial observations while blinded to the treatment and tumor implantation the samples in question had received, though it was largely immediately obvious if the animal belonged to a tumor-bearing cohort. Once observations were reported, the cohorts were revealed to the pathologist and a secondary analysis was conducted to probe how the trends observed from blinded samples aligned with the treatments administered. In general, tumors observed in this study are poorly differentiated adenocarcinomas with abortive gland formation and abundant solid areas, matching previous reports for this cell line. Several mice had nodular tumors growing in or adjacent to the pancreas, with minimal invasion of adjacent pancreatic parenchyma, likely due to leakage during the implantation procedure. This tumor also exhibited a tendency to coat serosal surfaces, particularly of the spleen with an often pronounced desmoplastic response and variable round cell infiltrates. The tumor showed little tendency to infiltrate the underlying spleen. However, in contrast to the usual situation with human pancreatic carcinomas, the mouse model failed to show perineural invasion or invasion into blood or lymphatic vasculature. It did, however, show a tendency to spread within the abdominal cavity on to the splenic surface, a common trait of pancreatic

adenocarcinoma in humans. Another common pattern of tumor deposition showed tumor cells infiltrating fat, as is common for pancreatic adenocarcinoma. Some of the tumor nodules exhibited central cavities of necrosis, particularly those treated with to the RGD- modified *C. novyi*-NT.

Inflammatory infiltrates including tumor-infiltrating lymphocytes were unevenly distributed through-out samples, ranging from virtually absent in PBS treated, tumor bearing cohort samples, to quite dense in the RGD-modified *C. novyi*-NT treated cohorts. The interface of tumor in pancreas tended to have a dense lymphoplasmacytic infiltrate as would be considered normal during early tumor development, with neutrophils an uneven feature–usually located in tumor adjacent to areas of cavitation. Of interest, a fairly uniform pattern of dense mononuclear infiltrate at the tumor-pancreas interface was observed that seemed to be escalated for RGD-modified and unmodified *C. novyi*-NT treated samples once unblinded to treatment. Small micro abscesses were a common feature of the fatty infiltration pattern. Variable lymphoid infiltrates were also a common pattern.

Due to *C. novyi*-NT species-specific DNA detection in heart, lung, and brain, slides from these organs were closely examined at least twice by the Board-certified pathologist/microbiologist (JMS). In keeping with other reports, the murine spleens generally contained large cells with atypical, multi-lobated nuclei that likely represent megakaryocytes/extramedullary hematopoiesis. None of the organs showed any cavities, abscesses, significant acute inflammation with neutrophils, or areas of necrosis. Occasional samples from each organ contained small nonspecific lymphoplasmacytic aggregates–which is common in most pathologic materials and no particular pattern was noted. Given the lack of any localizing features, higher power examination under oil of tissue gram stains was considered of no additional utility and likely would have more artifacts than any real findings. Morphologically, there was nothing to suggest damage to lung, heart, or brain occurred after exposure to the spores. All slides resulting from this study have been deposited in Zenodo, an open source data repository [33], and subsequently can be viewed at DOI: 10.5281/zenodo.8222869.

**Laser capture microdissection (LCMD).** Laser capture microdissection (LCMD) was conducted on the central necrotic regions of Fig 5A–5C, and differential PCR using species-

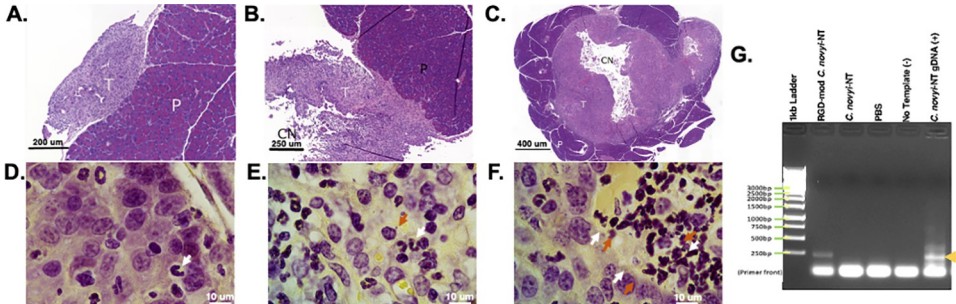

**Fig 5. Representative histology images and laser capture microdissection.** Representative images selected by Board-certified pathologist blinded to treatment from tumor implanted cohorts. Non-tumor implanted cohorts were analyzed and not found to have any signs of systemic inflammation. (A-C) H&E staining at 20X of (A) Representative image selected from the tumor bearing, PBS treated cohort, (B) Representative image selected from the tumor bearing, *C. novyi*-NT treated cohort, (C) Representative image selected from the tumor bearing, RGD-modified *C. novyi*-NT treated cohort. D-F) Brown-Hopps Gram Staining at 100X staining of (D) Representative image selected from the tumor bearing, PBS treated cohort, (E) Representative image selected from the tumor bearing, *C. novyi*-NT treated cohort, (F) Representative image selected from the tumor bearing, RGD-modified *C. novyi*-NT treated cohort. All slides from this study have been deposited in Zenodo (DOI: 10.5281/zenodo.8222870). *Legend*: CN-central necrosis, T-tumor, P-Pancreas, Arrows: Orange—*C. novyi* 2μm spore, White–neutrophils. (G) Laser Capture Microdissection of the central necrotic (CN) regions identified in A-C on immediately contiguous slides. Re-isolated samples underwent subsequent differential PCR using species-specific *C. novyi*-NT 16S rRNA-targeted primers.

specific *C. novyi* 16S rRNA primers with normalized template DNA concentration was conducted. A signal resulted from the DNA isolated from the slide representing the tumor bearing, RGD-modified *C. novyi*-NT cohort sample, but neither of the tumor bearing *C. novyi*-NT or PBS treated cohort representative samples indicated the presence of *C. novyi* DNA (Fig 5G). The presence of primer fronts is a common characteristic of these primers since their original design and publication [15, 17, 18]. Previously confirmed *C. novyi*-NT genomic DNA [20] was used as a positive control, while no template (volume made up with molecular grade water) was used as a negative control.

**Multiplex immunofluorescence (MxIF).** Since the histopathological analysis of all slides indicated unique inflammation in the pancreata and associated tumor slides of the tumor bearing, RGD-modified *C. novyi*-NT cohort (as represented by the selected images in Fig 5C and 5F), and *C. novyi*-NT DNA presence was confirmed (Fig 5G); multiplex immunofluorescence [34] was performed on a consecutive slide to further detail the influence of RGD-modified *C. novyi*-NT upon a localized immune response (Fig 6). DAPI-stained nuclei were referenced to correctly overlay multiple antibody signals. Mucin-1 (MUC1) is overexpressed on the surface of pancreatic tumor cells [35–37], and thus indicated tumor cells present in the sample (Fig 6C, 6G and 6H). Myeloperoxidase (MPO), was employed as a marker of neutrophil presence [38, 39] and implies an inflammatory context as shown (Fig 6E and 6J). Further, F4/80 is a well characterized cell surface marker for mature murine macrophages [40] and was used to detail the presence of these cells in response to RGD-modified *C. novyi*-NT presence within the tumor microenvironment (Fig 6D and 6I). While an antibody has been reported to be raised against *C. novyi*-NT elsewhere in the literature [15, 19], no such antibodies have been thoroughly detailed in the literature, or made widely available. When these immunofluorescence markers were overlaid with where *C. novyi* species-specific 16S rRNA-encoding DNA was isolated from on a consecutive slide, several colocalization loci were observed (Fig 6A–6F), indicating macrophage and neutrophil recruitment to the pancreatic tumor, as has been previously demonstrated to be characteristic of *C. novyi*-NT tumor localization [15]. This is of particular promise as pancreatic tumors are known to be immunologically 'cold' tumors with very little immune stimulation observed [9].

## Discussion

Oncolytic bacteria, and in particular *Clostridium novyi* with its characteristic biphasic life cycle that requires specific environmental conditions [17, 20], demonstrate great promise for

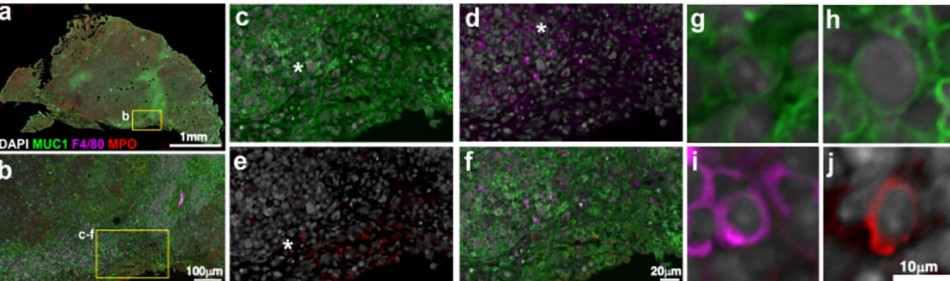

**Fig 6. Effects of RGD-modified *C. novyi*-NT on localized immune response.** Multiplex Immunofluorescence staining of RGD-mod CnNT representative sample from different staining rounds with mesoscale imaging aligned and merged. (A) Overview of pancreas sample with evidence of cancer. Markers used in all panels: DAPI—gray, mucin 1 (MUC1)—green, F4/80 –magenta, myeloperoxidase (MPO)- red. (B) Region taken from the immune-active region in the tumor. (C-F) Region from B, indicates areas of macrophages (D) and neutrophils (E) near mucin-positive cells. (G-J) Cellular resolution of myeloid cells and putative areas of tumor from areas indicated by asterisks (*).

treating solid tumors—especially in cancers that current therapeutics have demonstrated limited efficacy. While clinical trials are currently underway to advance direct intra-tumoral injections of *C. novyi*-NT spores [24, 41], pre-clinical trials probing the efficacy of this same bacteria delivered intravenously demonstrated a very high clearance rate [16, 19] that would limit widespread translation. This study attempted to overcome this pitfall using CRISPR-mediated genetic engineering (Supporting Information) to create a stable genetic construct [20] that allowed *C. novyi*-NT to accomplish a higher rate of tumor localization in a pancreatic tumor model.

## Characterization of RGD-modified *C. novyi* spores

Determining the presence and functional ability of the expressed RGD peptide to bind with the $\alpha_V\beta_3$ integrin, which is over-expressed on pancreatic tumor cells, was very challenging as many of the traditional immunohistochemistry techniques are not available due to a lack of RGD antibodies. Furthermore, there is very little to no discussion in the published literature detailing the presence of $\alpha_V\beta_3$ integrin or RGD analog expression in bacteria of any species—nonetheless spores. Hence, in order to evaluate the physical presence and functionality of the RGD-encoding sequence genetically inserted into the *C. novyi* genome, a novel adaptation of an adhesion assay [30] to assess the interaction between RGD and its integrin binding partner was created, marrying the fields of general bacterial adhesion assays and what has been published describing the interaction in adhesion assays between human RGD and $\alpha_V\beta_3$ (Fig 1A). Importantly, before treating an integrin coated surface with *C. novyi* spores, X-ray photoelectron spectroscopy (XPS) (Supporting Information) confirmed the presence of integrin on the borosilicate surface. While this assay was able to functionally demonstrate modified *C. novyi* spores showed preferential adhesion to the integrin coated surface (Fig 1B and 1C), it was unable to directly demonstrate the expression of RGD on the spore surface.

The spore coat of *C. novyi* is characterized by several layers, including (from outer most to inner most): sacculus, amorphous shell, and honeycomb layers [32] (Fig 2A). Beyond the honeycomb layer there can be 3–6 layers consisting of an undercoat, cortex, germ cell wall, and the spore core. Several landmarks detailing spore coat architecture that had been previously published [32] were successfully observed. The dark, dense core (Co) and surrounding cortex (Cx) layers were observable, as was the gray layer just outside the cortex indicating the characteristic 5–7 layer coat (Ct) containing the dark staining granular (G) paracrystalline layer (Fig 2B). Certain spore coat layers (*i.e.*, an intact amorphous layer and the germ cell wall) were not observable due to both the method of isolation, which was selected to be both scalable and compatible for clinical spore preparation, as well as the TEM sample preparation.

While it is impossible to draw direct comparisons between TEM images due to the difference of angles used to slice the spores and variabilities in instrumentation, images generated of RGD-modified spores indicate a disorder in the paracrystalline layer, the target of RGD expression, when compared to both the unmodified *C. novyi* spores imaged in this study as well as in previous publications [32]. This study notably utilized a different method of preparation and thus the sacculus layer from previously published images is absent. A sucrose gradient was chosen as the method of isolation to mitigate any toxicity that might remain from reagents used to isolate *C. novyi* spores, keeping in mind scalability and cost mitigation. The loss of the sacculus layer through this isolation technique—being the commercially applicable and least toxic method—was anticipated and thus the method of genomic insertion of the RGD peptide targeted the honeycomb layers for expression instead. However, the core (Co), cortex (Cx) and other coat layers (Ct), as well as the dark granular layer (G) of the outer edge of the cortex can be seen. The dark granular layer (G) that represents the paracrystaline layer has been

previously published to be near the sacculus that this method of isolation removed, and thus this layer is much closer to the edge of the spores in our images. Yet, these images seem to indicate a fundamental difference in the spore coat architecture with a disrupted protein coat organization that seems correlated with the genetic incorporation of the RGD-encoding gene (all images are included in Supporting Information).

### *In Vivo* biodistribution of intravenously injected *C. novyi-NT* spores

This portion of the study was designed to establish the biodistribution of modified spores, and thus tail vein injection was used as the route of administration in the C57BL/6 mouse model as this is the most clinically relevant to establish the migration of RGD-modified *C. novyi*-NT spores from the blood stream into a solid pancreatic tumor in organisms with an intact immune system (Fig 3A and 3B). Both cohorts with pancreatic tumors and *C. novyi*-NT spore injections constituted a greater percentage of overall mouse mass than those that had PBS tail vein injections administered (Fig 3C). This, when taken in light of the data in Figs 5 and 6, likely represents immune cell recruitment and fluid retention due to inflammation from successful *C. novyi*-NT localization, as is corroborated by the bacterial burden data presented in Fig 4B. One of the primary benefits of using oncolytic bacteria is the combination of tumor cell lysis and immune cell activation [8, 17]. Furthermore, the reduction in percent mass upon administration of RGD-modified *C. novyi*-NT spores when compared to the non-modified spores may represent increased tumor lysis based on increased localization and subsequent earlier colonization as indicated by the bacterial burden established in Fig 4. It is possible that this increased bacterial burden could not only represent successful bacterial localization but also the beginnings of proliferation and thus colonization occurring within the tumor microenvironment. A previous study detailing tumor mitigation accomplished by intravenously delivered *C. novyi*-NT spores did not observe significant tumor mitigation until 2-3days post introduction [19], with full tumor lysis occurring at the earliest at 12 days post injection. The current study only allowed for a 24hrs post injection time point so as to capture the biodistribution of the spores, which had been reportedly cleared in less than 24hrs in other similar models [17, 19]. Notably, using our methods, the amount of time for *C. novyi*-NT cultures to double is around 24hrs [20] with RGD modification not causing an alteration of this growth curve [20].

Intravenous delivery of *C. novyi*-NT spores to a tumor has occurred before in the literature [19]. While this landmark study provided a substantial foundation for our study, it is critical to note that there are several differences that build upon the critical work of Diaz *et al*. First, the tumors implanted by Diaz *et al* were generated by subcutaneous injection of CT26 colon carcinoma cells into the flanks of BALB/c mice to probe the pharmacologic and toxicological characteristics of intravenous *C. novyi*-NT spore treatment. Conversely, we chose an orthotopic, pancreatic tumor model to establish the biodistribution of RGD-modified *C. novyi*-NT spores. It is crucial to draw two main comparisons between these studies detailing intravenous delivery of spores: 1) BALB/c mice have a notably higher rate of clearance than the immunocompetent C57BL/6 mice used in our study as well as a Th2 biased immune system [42], and 2) CT26 cells originate from a murine colon carcinoma and were used to create a non-organ based tumor, while our study used a murine pancreatic carcinoma cell line, KPC 5508, to generate syngeneic tumors within the pancreas. The Diaz *et al* study introduced $1.5 \times 10^{10}$ unmodified *C. novyi*-NT spores via tail vein injection at their highest dosage and demonstrated substantial tumor mitigation when less than a relative 1% of the total spore dosage reached the tumor after rapid clearance of the majority of the spores within 24hrs. After a single hour, Diaz *et al* demonstrated more than 80% of the relative initial injection had been cleared from the blood

stream and could not be located anywhere in the mouse as indicated by $^{125}$I-labeled spores. In the current study, a substantially lower initial dose of $1\times10^5$ unlabeled spores were injected via tail vein injection in a strain of mouse with an intact immune system. Even at this significantly lower dosing, the modified *C. novyi*-NT spores were able to localize to the tumor and showed preliminary indications of tumor mitigation, including immune cell activation and recruitment. This dose was determine to be in line with the dosages currently undergoing clinical trials for intratumoral injection [22–24] as well as calculated to avoid any potential for blood clotting caused by the addition of an RGD-peptide to the spore coat.

To further probe the efficacy of the *C. novyi*-NT spores with an RGD-modified spore coat, the burden of bacteria was quantified by PCR using primers specific for *C. novyi* 16S rRNA in normalized samples of genomic DNA. Quantification was performed using genomic DNA from tissue samples that had previously indicated the presence of *C. novyi*-NT (Fig 4A). When the ng/amplicon calculated based on the densitometry of the PCR is extrapolated to represent a relative percent of the average total *C. novyi*-NT DNA remaining in mice dosed with RGD-modified spores (86.65ng from Fig 4B), it was found that a relative 41% (35.34ng from Fig 4C) of the remaining bacterial load was successfully localized from the blood stream into the pancreas and associated tumor tissue (which could often not be clearly dissected away from the normal pancreatic tissue). When the same methods are applied to the data generated to quantify the bacterial burden of non-modified *C. novyi*-NT, 29% of the total bacterial burden was localized to the pancreas and associated tumor tissue. However, it should be noted that almost *twice* as much overall *C. novyi* DNA was found to be present in animals that received the RGD-modified spores over those that received wild-type spores. When this difference is accounted for, only a relative 14% of wild-type spores accomplished translocation to the pancreas and pancreatic tumor. Thus, through these calculations, the RGD modification resulted in not only twice the bioburden of spores throughout organs after 24hrs, but also an almost 30% increase in relative tumor localization to the pancreas and tumor.

While the presence of 16S rRNA is specific to *Clostridium novyi*, and thus adequate to demonstrate biodistribution of the bacterial species in general, it does not indicate whether the injected spores have (1) germinated to the vegetative phase, (2) remained as spores, or (3) have been phagocytosed. Furthermore, the mouse model studies reported here represent a pilot study to demonstrate the potential of pursuing such a path for the development of an intravenously capable *C. novyi*-NT oncotherapeutic. Thus, additional animals would need to be examined in a statistically powered cohort designed to determine the spore load in the pancreas and tumor; nonetheless, the RGD modification seems to have conferred not only increase in the overall bioburden after 24hrs—an indication of increased circulation—but also an increase in tumor localization when compared to previous studies [15, 19].

## Analysis of *in vivo* model samples

In contrast to the usual situation with human pancreatic carcinomas, the mouse model failed to show perineural invasion or invasion into blood or lymphatic vasculature. It did, however, show a tendency to spread within the abdominal cavity on to the splenic surface, as is common for pancreatic adenocarcinoma in humans. Some nodular tumors showed central cavitation/ necrosis, indicating tumor microenvironments necessary to adequately attract *C. novyi*-NT spores from the blood stream was achieved, despite the relatively small tumor size. Primary pancreatic adenocarcinomas in humans tend to be locally infiltrative and perineural and angiolymphatic invasion are common features as well. Notably, this model system did not recapitulate most of those features, however, serosal spread was observed. Metastatic disease was not found in any of the multiple organs sampled, nor observed when necropsies took

place. Of course, 14 days is a short time in the usual evolution of a tumor, and perhaps metastasis would develop later. Likewise, no evidence of systemic acute inflammatory response was observed after mice were exposed to spores intravenously. All slides resulting from this study have been deposited in Zenodo (DOI: 10.5281/zenodo.8222870) [33] and can be viewed within this repository.

With an intravenous injection occurring via tail vein, the expected first point of clearance would be the lungs. As possibly the most aerobic environment in the body, it would be surprising if the spores had germinated within the lungs and is likely that the DNA isolated from homogenization of the lungs is contained in the associated lymphatic tissue during normal clearance processes. Since *C. novyi*-NT species-specific DNA was detected in heart, lung, and brain (Fig 4C and 4D), slides corresponding to these samples were closely examined at least twice by the Board-certified pathologist/microbiologist (JMS). No abnormal cavities, abscesses, significant acute inflammation with neutrophils, or areas of necrosis were observed in these organs, supporting the hypothesis that the *C. novyi*-NT species-specific DNA detected was the result of phagocytotic waste or spores that were caught passing through local vasculature at the time of sample collection rather than off-target localization. Even if spores had localized to these areas, no bacterial-mediated inflammation was observed, indicating that germination from the immunologically inert spore form had not occurred. In this circumstance, previous literature indicates that clearance from the subject would occur within a reasonable timeline without issue [19].

While the presence of *C. novyi*-NT species specific DNA was detected in homogenized pancreas and associated tumor samples (Fig 4C and 4D) and supported by pathohistological analysis (Fig 5A–5F), these samples do not provide evidence regarding where *C. novyi*-NT is localized within these samples. To probe relative localization, laser capture microdissection was conducted on slides from the pathologist-selected representative samples. Specifically, the central necrotic regions were reisolated from de-paraffinized slide mounts, and differential PCR was conducted using species specific *C. novyi*-NT 16S rRNA-targeting primers. The resulting amplicons (Fig 5G) indicate that *C. novyi*-NT accomplished penetration into the central necrotic region of the tumor-bearing, RGD-modified *C. novyi*-NT treated cohort representative sample. Intriguingly, *C. novyi*-NT DNA was not amplified out of the tumor-bearing, *C. novyi*-NT treated cohort representative sample. This data supports the hypothesis that RGD-modification confers added localization to pancreatic tumors from intravenous administration over the innate localization ability of non-modified *C. novyi*-NT. Most importantly, it indicates that CRISPR/Cas9 modification of *C. novyi*-NT can be accomplished, giving rise to added anti-cancer capacities.

To further corroborate the presence of RGD-modified *C. novyi*-NT in the tumor microenvironment, multiplex immunofluorescence was used to probe the localized immune response observed by histopathological analysis (Fig 5E and 5F) and in previously published studies [15, 16, 43]. This study used MUC1 as a marker for pancreatic tumor cells [35–37], MPO as a marker for neutrophils [38, 39], and F4/80 to indicate murine macrophages [40]. Additionally, MUC1 has demonstrated physical interaction with HIF-1α, or hypoxia inducible factor 1 subunit alpha–a protein well characterized to function as a master regulator of hypoxia response-leading to stabilization at the protein level [36, 44]. Therefore, MUC1 immunofluorescence signaling can be extrapolated to represent HIF1α presence and subsequently hypoxia levels as well. While an antibody has been reported to detect *C. novyi*-NT [15, 19], these antibodies have not been thoroughly detailed in the literature regarding the antigen, or made widely available. It should be noted, particularly since the specific epitope of the reported antibody has not been published, it is possible that RGD-modification and/or culturing as well as isolation differences could prevent recognition even if an antibody was available. While *C. novyi*-NT

cannot be directly detected in Fig 6, the slide analyzed was isolated from the tumor-pancreas margin immediately contiguous to the sample in Fig 5, which indicated the presence of species-specific *C. novyi*-NT DNA. It is reasonable to therefore infer RGD-modified *C. novyi*-NT remains present in the center of the tumor just beyond the analyzed margins. When the resulting immunofluorescence signals are overlayed, co-localization of neutrophils and macrophages are found at the tumor margins indicating the beginning stages of immune recruitment and reactivation that are known to be characteristic of *C. novyi*-NT tumor colonization [15, 43]. This suggests that CRISPR/Cas9-mediated gene editing resulting in modification of the *C. novyi*-NT spore surface was accomplished while maintaining the potent immune re-activation [15, 43] that comprises an extraordinarily promising aspect of *C. novyi*-NT-mediated cancer therapeutics.

## Conclusions

The studies detailed in this manuscript demonstrate the use of a non-toxic, genetically modified *C. novyi*-NT bacteria to efficiently and effectively target and localize to a pancreatic tumor from intravenous administration—offering hope for a novel therapeutic to impact the grim five-year survival rate of a pancreatic cancer diagnosis. Additionally, this study is among the first to characterize the effects of CRISPR gene modification in harnessing the vast potential for bacterial pharmaceutical applications. Several species of bacteria, including *C. novyi*-NT [15–17, 19, 22], have demonstrated an innate ability to not only accomplish selective targeting of the microenvironment contained within a solid tumor, but also the colonization and eradication of the tumor to its margins [6–8, 13, 45]. Upon reaching the tumor margins, oncolytic bacterial therapy continues to demonstrate efficacy in actively recruiting and reactivating the immune system to accomplish both the eradication of the tumor margins as well as the bacterial abscess now contained within the tumor [6, 8, 13, 15, 16]. Recent technological advances (*e.g.*, CRISPR/Cas gene modification techniques) have made it possible to pursue the design and development of genetically modified forms of these oncolytic bacteria in order to confer further advantages while honing their innate tumor eradication abilities [7]. However, a large knowledge gap remains surrounding the basic biochemical methods through which species detect and accomplish localization to the hypoxic core of a solid tumor, and only a skeletal knowledge has been obtained detailing how tumor lysis is accomplished. Furthermore, the methodology with which to establish the efficacy of these treatments remains to be fully developed. The use of a biologically living therapeutic overcomes many of the hurdles of current oncotherapeutics, including no longer being limited by the initial dosage since oncolytic bacteria can proliferate upon successfully colonizing the tumor. However, this ability to multiply confounds many of the modern pharmacological methods commonly used to determine efficacy and of course leads to a different potential risk profile.

While there are several hurdles to overcome in order to accomplish the clinical translation of intravenously delivered *C. novyi*-NT spores as an oncotherapy, the potential of this therapy should not be ignored. *C. novyi*-NT -mediated cancer therapeutics target attributes nearly ubiquitous to all solid tumors; potently activate the immune system at tumor margins; and ultimately enact tumor lysis directly, or, in the future, could deliver agents to hypoxic areas of tumors. A *C. novyi*-NT-based, IV therapy could target solid tumors regardless of tumor dormancy, recurrence, resistance, or metastasis either in a monotherapeutic approach, to complement and synergize with current therapies, or as a delivery vehicle. Further, *C. novyi*-NT is capable of biological replication, reducing production cost and addressing socioeconomic concerns. *C. novyi*-NT therapy would not require expensive, time-consuming tumor characterization prior to treatment and can be used for tumors composed of multiple cell types, including

stem and resistant cells. Finally, spores are stable in almost any environment, circumventing expensive storage requirements, and allowing for distribution into rural and poor communities.

Challenges as the field progresses toward clinical translation cannot be ignored, even at these early stages, and will include developing the necessary methodology to produce spores that are consistent in composition across multiple batches as well as scaling up to large batch production [7]. Patient and physician consent may also prove to be a challenge as injecting bacteria into a patient's veins is far from intuitive in terms of treatment for any disease state [7]. Nevertheless, this study demonstrates two things; first, that oncolytic bacteria and in particular *Clostridium novyi*-NT can be genetically modified to accentuate their innate tumor mitigation, and second, that RGD-modified *C. novyi*-NT can successfully localize to a tumor with a much smaller dose than previously used in published studies and still illicit its characteristic immune activation. We believe this study clearly demonstrates that there is great potential in applying CRISPR/Cas-mediated genetic engineering techniques in the quest for a better oncotherapeutic, particularly in the face of such a potent threat as pancreatic cancer.

## Materials and methods

### Vertebrate animals ethics statement

The animal studies reported in this manuscript were approved by North Dakota State University's IACUC prior to initiation as protocol #A21003 under Primary Investigators AEB and SM. As detailed below, to ensure humane practices 1–3% oxygenated isoflurane was used both prior to and during surgical procedures as an anesthetic. Isoflurane overdose was the method of euthanasia employed, with cervical dislocation to ensure sacrifice prior to necropsies.

### Generation and preparation of RGD-modified *C. novyi* spores

**Bacterial culture.** *Clostridium novyi* was purchased from ATCC (19402) and grown up anaerobically in reinforced clostridial medium (RCM, BD Difco) supplemented with the oxygen fixing enzyme Oxyrase for Broth (Oxyrase, Inc) as previously described [20]. Sporulation of *C. novyi* and RGD modified *C. novyi* spores was accomplished according to a previously published protocol [19]. Briefly, a specialized growth medium replicating conditions that would force sporulation naturally was prepared and sterilized. Vegetative cells were inoculated into this medium and allowed to sporulate for seven days. Subsequent purification occurred via a protocol previously described for the isolation of *Clostridioides difficile* spores [46]. Resulting spores were quantified by serial dilution and enumerated on solid media containing Oxyrase for Agar as per manufacturer's instructions to normalize the concentration through calculating the CFUs/ml mathematically.

**CRISPR plasmid construction.** A plasmid containing a Cas9nickase enzyme, previously published to have successfully cleaved DNA in Clostridial species [47], was utilized to modify the genome of *C. novyi* cells (pKMD002) as previously published [20]. Briefly, to generate pKMD002, the sgRNA within the commercial pNICKclos1.0 plasmid, which contained the Cas9nickase enzyme, was removed through restriction digest using *Spe*I and *Not*I, replacing it with a sgRNA sequence that correlated with the putative [48] *C. novyi* spore coat protein encoding gene NT01CX0401. The sequences of pNICKClos1.0 containing homologous recombination domains were also exchanged via restriction digest (*Not*I and *Xho*I) with 1kb sequences corresponding to both upstream and downstream genomic sites targeted by the NT01CX0401 sgRNA. Within these two HDR arms, the genetic sequence for RGD was cloned. This insertion sequence also contained a start codon, Shine-Dalgarno sequences, and TATA box as well as a flexible GGA linker to allow for some molecular rotation of the RGD peptide

[20] (S1 Fig). All sequences were modified to account for species specific codon bias [20] (S1 Table). Additionally, an *EcoR*V restriction digest site was inserted into the HDR sequences to allow for verification of a successful genomic insertion.

**Transformation.**   Calcium competent *C. novyi* cells were created by modifying a standard protocol for *E. coli* [49] to include Oxyrase for Broth at 10% in calcium chloride solutions [20]. Resulting *C. novyi* cells were then transformed with 5μg of purified pKMD002 plasmid DNA via heat shock. After 24hrs, the resulting bacterial cultures were exposed to the selective marker, erythromycin (250μg/ml final concentration, Sigma Aldrich), for 24hrs at 37˚C anaerobically. Cells were then plated onto solid media containing reinforced clostridial media (RCM, BD Difco) with 10% Oxyrase for Agar (Oxyrase Inc.), erythromycin (250μg/ml) and 3% agar (Sigma-Aldrich) and grown in OxyPLUS anaerobic chamber plates (Oxyrase, Inc) for 48hrs. Candidate colonies were picked from the agar plates and grown in RCM broth containing 10% Oxyrase for Broth.

**Genomic insertion confirmation.**   Resulting candidate spore cultures underwent TRIzol (Zymo Research) genomic DNA isolation according to the manufacturer's protocol. The resulting genomic DNA then underwent PCR utilizing primers specific to the HDR domain (S1 Table) contained within pKMD002 which included the *EcoR*V restriction site as designed for verification. GoTaq Green PCR MasterMix (Promega) and nuclease free water (Promega) was used to perform PCR. The thermocycler program was as follows: 95˚C for 5min, (95˚C for 30sec, 50˚C for 1min, 68˚C for 2min) x40 cycles, and then 68˚C for 5min. The resulting oligonucleotides were then digested with *EcoR*V-HF enzyme (New England BioLabs, Inc) in CutSmart Buffer at 37˚C for 1hr. After digestion, samples were electrophoresed in a 1% agarose gel at 120V, then imaged [20] and analyzed to verify if genomic insertion had occurred after transformation with pKMD002 (S2 Table).

**Alpha toxin removal.**   A previously published method [17] was modified to include an activation step consisting of a 20min incubation at 55˚C after the membrane permeabilization incubation at 70˚C [20]. PCR screening was then conducted with primers [17] corresponding to *C. novyi* α-toxin in order to confirm knock-out [20]. Knock-out was confirmed immediately prior to murine introduction to ensure animal welfare and appropriate use (S4 Fig).

## Characterization of RGD-modified *C. novyi* spores

**Adhesion assay.**   A published protocol used to characterize RGD-modified exosomes [30] was adapted as follows. Circular borosilicate glass cover slips (Fisher Scientific) were treated with air plasma (corona) treatment (Enercon Compak 2000 Corona Treater Model LM4045-06) to make the surface charged and therefore reactive. Resulting cover slips were then placed into the center of a well of a 6-well plate (Celltreat) for further experimentation. The authors note that it is important for subsequent steps that the slide covers be in the center of the wells and not touching the sides. Purified $\alpha_V\beta_3$ integrin (100μL of 10μg/mL carrier-free, human recombinant protein, R&D Systems Bio-Techne 3050-AV) solution was administered to the center of the corona-treated cover slips. The optimal concentration of integrin was determined by employing X-ray photoelectron spectroscopy (XPS, also known as electron spectroscopy for chemical analysis or ESCA) to analyze coverslips incubated with a titration of purified $\alpha_V\beta_3$ integrin. When the amount of nitrogen detected on the coverslip no longer increased, it was interpreted to represent the maximum concentration of protein necessary to cover the entire surface as nitrogen is not a component of any reagent present other than the purified $\alpha_V\beta_3$ integrin (S4 Table). The protein carrier solution (1x phosphate buffered saline, VWR) was allowed to evaporate at 4˚C for 48hr, thus allowing deposition and subsequent adhesion of $\alpha_v\beta_3$ integrin to the cover slip. Once the carrier solution had evaporated, 100μL of purified *C.*

*novyi* spores normalized via spore enumeration to 1,000 spores/μL were applied. The 6-well plates were then returned to 4˚C to again evaporate the carrier solution (1x PBS), thus facilitating the adherence of the *C. novyi* spores to the surface. Slides were then rinsed with 1x PBS to remove any excess unadhered spores. 1% v/v crystal violet (CV) was then applied to the center of the coverslips to stain remaining spores adhered to the integrin coated surface. CV stain was allowed to incubate for 3hrs at room temperature. Excess CV stain was removed by three washes using 10% v/v acetic acid incubated on the cover slip for 10min while shaking on an orbital shaker at 150rpm. The resulting run off of acetic acid and crystal violet dye was removed to a 96 well plate and the absorbance at 590nm was observed and recorded. After all washes, coverslips with adhered, CV-dyed spores were then placed on slides for microscopy imaging. Brightfield confocal scanning microscopy images were obtained for each slide (Zeiss Axio Observer Z1 LSM 700). Quantification of blue/violet pixels on each slide was determined using FIJI open source image processing suite [31] on captured images at 4X magnification covering the entirety of each slide coverslip (S2 Fig). The image was separated into color channels, the area selected, and the corrected total cell blue/violet color (CTCC) was determined using the internal density and the area and mean blue violet color (Eq 1). A one-way ANOVA test ascertained statistical significance.

$$CTCC = \text{Integrated Density} - (\text{Area} * \text{Mean}) \tag{1}$$

**Transmission electron imaging (TEM).**   Specimens for transmission electron microscopy were fixed in 2.5% v/v glutaraldehyde in a 0.1 M sodium phosphate buffer, pH 7.35 (Tousimis Research Corporation, Rockville, MD) for at least 2hrs at 4μ. Specimens were rinsed twice in sodium phosphate buffer and then placed in 2% osmium tetroxide in buffer for 2hrs at room temperature. Following buffer rinse, water rinse, and dehydration in a graded acetone series, samples were embedded in Epon-Araldite-DDSA with DMP-30 accelerator and sectioned at 60-80nm thickness on an RMC MT XL ultramicrotome (Boeckeler Instruments, Tucson, AZ). Sections on copper grids were stained with lead citrate for two minutes and dried before being observed and photographed on a JEOL JEM-100CX II electron microscope (JEOL Inc., Peabody, MA).

## Immunocompetent mouse model

Wild type C57BL/6 mice were used in this pilot study. KPC 5508 pancreatic tumor cells (originally isolated from KPC mice) were injected as described below to create a tumor (NDSU IACUC #A21003). As a control, one cohort of mice underwent a sham surgery, in which sterile phosphate buffered saline (1x PBS) was injected in lieu of cells in order to observe the immune reaction due to surgical procedures alone. Mice were randomly assigned to treatment groups, with both male and female mice utilized (Fig 3B).

**Xenographic tumor implantation surgery.**   Mice were anesthetized by isoflurane (3% in 1 L/min 100% oxygen for induction, 2% in 1 L/min 100% oxygen for maintenance), and a lack of pedal reflex was obtained to assess anesthetic depth. Eye gel (Hanna Pharmaceutical Artificial Tears Ophthalmic Ointment) was applied to both eyes. Surgeries were performed on a sterile benchtop, and the animal was laid on a heating pad covered by a sterile drape to maintain body temperature and prevent hypothermia for the duration of the surgery. The fur of approximately a square inch surrounding the incision site on the left flank was removed by shaving. Betadine solution was then applied with sterile gauze in a circular fashion starting at the surgical incision site and rotating outward. The spleen of the mouse was visualized, and the abdomen of the mouse was then opened using sterile surgical scissors to create a 1cm

incision in the medial upper abdomen, just over the location of the spleen. KPC 5508 cells ($10^5$) were suspended in 25μL of sterile saline and loaded into a sterile syringe (28-gauge needle on a 0.5mL insulin syringe) for the tumor implantation cohort. For the mock tumor implantation cohort, 25μL of sterile PBS was loaded into a sterile syringe (28-gauge needle on a 0.5mL insulin syringe). The peritoneum was gently grasped with forceps and a small incision was made just over the spleen (~1cm). The spleen was grasped gently with sterile forceps, exteriorizing it along with the pancreas, and a small portion of the intestine. While holding the spleen vertically, the dark line indicative of the pancreatic vein running down the pancreas was located as a landmark for injection. The needle of the syringe was pre-loaded with either tumorigenic cells or PBS and then inserted parallel to the pancreatic vein, then the solution was slowly injected so that a small bubble appeared in the pancreas. The needle was then gently removed from the pancreas, and forceps were used to manipulate around either side of the opening in the peritoneum and gently return the organ into the peritoneal cavity. Subsequently, the peritoneum and skin were sutured in layers (Ethilon chromic gut, 5–0, 1.5 metric, 687G or Ethilon, nylon suture/black monofilament, 5–0, 1.0 metric, 698H). Neosporin and subsequently tissue glue were placed over the incision to prevent infection and reopening of the incision. The mouse was given buprenorphine (0.1mg/kg) subcutaneously for pain control. Mice were returned to micro-isolator housing after they had recovered normal posture and were walking around the cage freely/normally. Mice were then singly housed after surgery to help prevent suture disruption and monitored twice a day with Neosporin treatment of open wounds (*e.g.*, skin picking, pulled stitches) halting a week prior to treatment. Injected cells were allowed to develop into a tumor for 14days following implantation, with tumor progression being monitored daily by visualization and palpating the area every 3-5days.

**Tail vein treatment.**　Both mice bearing tumors and non-tumor bearing mice (those injected with shams of sterile saline) underwent treatment with purified *Clostridium novyi*-NT spores. Regardless of their tumor status, all cohorts underwent one of three treatments: 1) tail vein injections of 200μL sterile 1x PBS, 2) tail vein injections of $10^5$ CFUs *C. novyi*-NT spores in 200μL sterile 1x PBS, and 3) tail vein injections of $10^5$ CFUs RGD-modified *C. novyi*-NT spores in 200μL sterile 1x PBS. A table detailing the cohort specificities has been included for clarity (Fig 3B). All treatments were administered by a single tail vein injection (200μL) 14days post-surgery. Mice were anesthetized with isoflurane as previously described prior to tail vein injections. As a method to prevent hypovolemia and sepsis, which are rare complications following IV injection with attenuated bacteria [19], mice were given an injection (26-gauge needle on a 0.5mL syringe) of 200μL 37˚C sterile saline subcutaneously (SC) immediately following injection of *C. novyi*-NT spores. Mice received additional 200μL SC injections of sterile saline every 4hrs for a total of 1.2mL of saline per mouse.

**Euthanasia.**　Twenty-four hours following treatment with spores (or PBS in the control groups), mice were euthanized by isoflurane gas (5% in 2L/min 100% oxygen) and cervical dislocation. The spleen, pancreas, and associated tumor (the tumor was often inextricable from the pancreas), liver, kidney, lung, heart, and brain were then aseptically harvested with half of each organ being flash frozen in liquid nitrogen to undergo PCR for the presence of *C. novyi*-NT spores and the other half being submerged in 10% neutral buffered formalin (NBF) to undergo histological processing. Before being halved, the pancreata and associated tumor tissue were weighed to establish the mass.

## Characterization of RGD-modified *C. novyi-NT* spores *In Vivo*

**Homogenization of murine tissues.**　To homogenize harvested organs, a liquid nitrogen physical pulverization method was modified [50]. A brass hose connector and coordinating

cap were sterilized and chilled in liquid nitrogen. This hardware was then used to create a pulverization chamber in which the tissue and liquid nitrogen could be placed. A snug-fitting zinc-platted carriage bolt that corresponded to the hose connector was used as a piston that when tapped with a hammer pulverized the tissue. The resulting pulverized tissue was removed and placed in tubes that contained 0.5mm silica homogenization beads (BeadBug, Sigma Aldrich). Tubes were vortexed for 20min or until the tissue was sufficiently homogenized. This solution was then pelleted at 12,000rpm for 5min and the resulting supernatant was removed to another tube. Genomic DNA was then isolated via manufacturer's protocol for TRIzol (Zymo Research, Inc.).

**Biodistribution of *C. novyi*-NT spores.** The harvested genomic DNA underwent PCR with primers specific to 16S rRNA [17] characteristic of *C. novyi* to determine the presence or absence of spores in the organs harvested. Genomic preparations (5μL) were loaded as the template with GoTaq Green PCR MasterMix (Promega), nuclease free water (Promega), and 10μM final concentration of primers. The thermocycler program was as follows: 95˚C for 5min, (95˚C for 30sec, 50˚C for 1min, 68˚C for 2min)x40 cycles, and then extension at 68˚C for 5min. Resulting amplicons were run on a 1% agarose gel at 130V and imaged.

**C. novyi-NT bacterial burden quantification.** The genomic DNA for samples that resulted in an amplicon after PCR was quantified to obtain the concentration (ng/μL) and quality (Abs 260/280) of the extracted sample (S5–S11 Figs). The resulting quantifications were used to calculate a corrected concentration that accounted for samples with low quality (Eq 2). The corrected concentration (ng/μL) was then used to normalize all samples to 19ng per reaction. Subsequent PCR occurred with 16S rRNA primers using the same thermocycler program and gel procedure as detailed previously. The resulting amplicons were quantified using FIJI software to create an intensity plot for each band on the gel and then quantifying the area contained within the peak that corresponded to the 16S rRNA band (S5 Table, S12 Fig). This area was then normalized to allow for comparison between multiple gels by utilizing the known quantification of the 3kb band contained in the 1kb molecular ladder (TriDye 1kb Ladder, Promega) to convert the area value to nanograms of DNA contained in the amplicon.

$$\text{Corrected ng/μL} = (\text{ng/μL}) \times (260/280) \tag{2}$$

## Histological analysis

**Slide mounts and staining.** Formalin-fixed tissue samples were processed for paraffin embedding by the means of dehydration, clearing and paraffin infiltration (Lynx II Tissue Processor). The paraffin embedded tissue samples were sectioned using a Leica Rotatory Microtome RM2125 RTS at 5μm thickness. Designated tissue sections were subsequently deparaffined and stained with H&E and gram staining as per standard histology protocols (Leica Autostainer XL). Whole slide scanning (WSI) was performed by a Panoramic 250 whole slide scanner at 20X magnification (3D Histech) using a Carl-Zeiss Plan-Apochromat 20X / NA 0.8 objective. Gram-stained slides indicated as of interest by the pathologist were further imaged using oil emersion on a Nikon Eclipse Ti2 at 100X magnification and reviewed further by the pathologist.

**Representative image/sample selection.** Digital, whole slide scanned H&E and gram-stained images of major organs from murine subjects were reviewed in duplicate by a qualified and Board-certified expert in infectious disease pathology (James M. Small, MD/PhD) using the opensource digital pathology software, QuPath. Representative images and samples indicative of the whole cohort were selected, exported as images, and annotated by the pathologist who was blinded to the treatment each cohort underwent until after representative samples were selected.

**Laser capture microdissection.** Slide mounts immediately consecutive to those examined by the pathologist were made for pancreatic and associated tumor samples by the UNMC Tissue Sciences Core Facility. Samples were deparaffinized and stained with H&E using standard protocols. A Zeiss PALM MicroBeam IV laser capture microdissection system mounted on a Zeiss AxioObserver microscope with motorized stage and Qiagen Micro DNA Amplification kit was used to remove cells from the center of tumors up to the tumor margins per manufacturer's protocols. Differential 16S rRNA PCR was conducted on the resulting samples using GoTaq Green PCR MasterMix (Promega), nuclease free water (Promega), and 10μM final concentration of primers [17, 20]. The thermocycler program was as follows: 95°C for 5min, (95°C for 30sec, 50°C for 1min, 68°C for 2min)x40 cycles, and then extension at 68°C for 5min. Resulting amplicons were run on a 1% agarose gel at 130V and imaged.

**Multiplex immunofluorescence (MxIF).** Formalin fixed, paraffin imbedded slides were deparaffinized in xylene and rehydrated through an ethanol gradient as per standard protocol [34]. Following rehydration, antigen retrieval was performed through heating acidic citrate buffer just below boiling (20min, low heat, microwave) and an additional benchtop, room temperature incubation (20min). Slides were blocked in 1% BSA for 1hr and then permeabilized in 1.5% triton for 10min. The slides were then stained with antibodies that were either directly conjugated to fluorophores or a separate, unconjugated primary and secondary antibody when appropriate as to not risk cross reaction with the other antibodies in the panel. Specifically, the MPO antibody (Abcam, ab252131) was purchased pre-conjugated, but the MUC1 (Thermofisher, MA5-11202) and F4/80 (Novus, NB600-404) antibodies were conjugated in-house using a Cy5 (Amersham CyDye kits, GE Health) and a Cy7 labeling kit (PE/Cy7 Conjugation Kit—Lightning-Link, Abcam) respectively (S4 Table). When BSA was present in the manufacturer's formulation for the antibody solution, it was removed prior to labeling using a purification kit (Thermo Fisher Scientific, Melon Gel IgG Spin Purification Kit). Coverslips were then secured to the slides using mounting media with DAPI (Prolong Gold Antifade Mountant, ThermoFisher Scientific) and allowed to dry at least 48hr. Whole slide scanning was performed by a Panoramic 250 whole slide scanner at 20X magnification (3D Histech) using a Carl-Zeiss Plan-Apochromat 20X / NA 0.8 objective. Fluorescence was excited by a SPECTRA X light engine (Lumencore) through Dapi/Cy2/Cy3/Cy5 a Zeiss quadband filter cube (Shemrock: LF405/488/561/635-B-000) and the Cy7 channel was excited using a Bright-Line filter set (Shemrock: Cy7-B-000).

## Statistical information

For statistically relevant sample sizes (Fig 2B and 2C), a one-way ANOVA test ascertained statistical significance using an F value of 2.93 for the 0.05 significance level. In Fig 2B, * denotes a *p* value of < 0.05, while in Fig 2C, ‡ denotes a *p* value equal to 0.05. This experiment was repeated three times with an n of 6 per replicate for a total n of 18 and data are presented as the cumulative mean of all data points gathered. All statistics were done using SigmaPlot software and confirmed by the Center for Diagnostics and Therapeutic Strategies in Pancreatic Cancer Biostatistics Core Facility (MO). For samples where statistical significance could not be ascertained due to the pilot nature of this study, error bars represent standard deviation from the mean. No cohorts are reported with an n of less than 3.

## Supporting information

**S1 Table. DNA sequences relevant to the cloning and confirmation of the pKMD002 plasmid.**
(DOCX)

**S2 Table. The results of CRISPR/Cas9n mediated genomic insertion of the RGD peptide into *C. novyi* candidates A-E and controls.** (summarized from previously published work ref 20).
(DOCX)

**S3 Table. Assessing the $\alpha_V\beta_3$ coated surface designed for an adhesion assay.** X-ray photoelectron spectroscopy was conducted to assess the elements present on a borosilicate sild cover after corona plasma treatment and subsequent coating with integrin. Adequate presence of integrin was determined when silicate was no longer detectable and nitrogen content had reached its maximum.
(DOCX)

**S4 Table. Antibodies used for multiplex immunofluorescence staining of mouse model samples in Fig 6.**
(DOCX)

**S5 Table. Key to the samples run in S12 Fig.**
(DOCX)

**S1 Fig. CRISPR-mediated genomic insertion of RGD peptide encoding gene.** Schematic representation of the CRISPR cloning cassette utilized in pKMD002 for gene insertion.
(DOCX)

**S2 Fig. Representative images used to generate the CV pixel count quantification of the adhesion assay.**
(DOCX)

**S3 Fig. TEM images of unmodified *C. novyi* NT spores.**
(DOCX)

**S4 Fig. TEM images of RGD-modified *C. novyi* NT spores.**
(DOCX)

**S5 Fig. Generation of a non-toxic RGD-modified *C. novyi* spore.** In order to accomplish *in vivo* introduction without substantial toxicity, the α-toxin encoded phage DNA had to be knocked out in *C. novyi* that had already undergone successful genetic modification with RGD-encoding DNA. Upon knockout, PCR was conducted with primers specific to the α-toxin so that a lack of a band around 500bp represents a-toxin removal.
(DOCX)

**S6 Fig. 16S rRNA PCR amplicons used to establish the biodistribution for the mock tumor (PBS) implant cohort treated with PBS via tail vein injection.** (Sp–spleen, P–pancreas, Lv- liver, Kd- kidney, Lu- lung, Ht- heart, Br- brain, (-) no template control, *E. coli* DNA control).
(DOCX)

**S7 Fig. 16S rRNA PCR amplicons used to establish the biodistribution for the mock tumor (PBS) implant cohort treated with *C. novyi* NT spores (100,000) via tail vein injection.** (Sp–spleen, P–pancreas, Lv- liver, Kd- kidney, Lu- lung, Ht- heart, Br- brain, (-) no template control, *E. coli* DNA control).
(DOCX)

**S8 Fig. 16S rRNA PCR amplicons used to establish the biodistribution for the mock tumor (PBS) implant cohort treated with RGD-modified *C. novyi* NT spores (100,000) via tail**

**vein injection.** (Sp–spleen, P–pancreas, Lv- liver, Kd- kidney, Lu- lung, Ht- heart, Br- brain, (-) no template control, *E. coli* DNA control).
(DOCX)

**S9 Fig. 16S rRNA PCR amplicons used to establish the biodistribution for the tumor (KPC) implant cohort treated with PBS via tail vein injection.** (Sp–spleen, P–pancreas, Pt–pancreatic tumor, Lv- liver, Kd- kidney, Lu- lung, Ht- heart, Br- brain, (-) no template control, *E. coli* DNA control).
(DOCX)

**S10 Fig. 16S rRNA PCR amplicons used to establish the biodistribution for the tumor (KPC) implant cohort treated with *C. novyi* NT spores (100,000) via tail vein injection.** (Sp–spleen, P–pancreas, Pt–pancreatic tumor, Lv- liver, Kd- kidney, Lu- lung, Ht- heart, Br- brain, (-) no template control, E. coli DNA control).
(DOCX)

**S11 Fig. 16S rRNA PCR amplicons used to establish the biodistribution for the tumor (KPC) implant cohort treated with RGD-modified *C. novyi* NT spores (100,000) via tail vein injection.** (Sp–spleen, P–pancreas, Pt–pancreatic tumor, Lv- liver, Kd- kidney, Lu- lung, Ht- heart, Br- brain, (-) no template control, *E. coli* DNA control).
(DOCX)

**S12 Fig. 16S rRNA PCR amplicons from normalized genomic DNA isolates used to establish the bacterial burden in ng/amplicon in Fig 6. The sample legend can be found in S5 Table.**
(DOCX)

## Acknowledgments

Special thanks to the NDSU Agriculture Department's Advanced Imaging Microscopy Laboratory, specifically Dr. Pawel Borowitz and Jordan Flaten, as well as the Electron Microscopy Core for TEM studies, and the Research Operations Recharge Center and Characterization Service Center for XPS analysis. Further thanks to the UNMC Tissue Sciences Facility as well as the UND Histology Core for their services in preparing the histology slides within this publication.

We would like to acknowledge the use of Biorender.com for the creation of several figures contained within this publication. Finally, we thank our NDSU colleagues, specifically Jacob W. Shreffler, Alexandro Delgado, Reed I Jacobson, Paige R. Johnson, and Dr. Jiha Kim for their contributions to this study and publication as well as our UNMC colleagues, specifically Thomas C. Caffery, Kelly A. O'Connell, Aleata A. Triplett, Jennifer L. Endres, William R. Miklavcic, Kyle L. McAndrews, Dr. Heather Jensen Smith, Dr. Stacy Gilk, and Dr. Leah Cook.

## Author Contributions

**Conceptualization:** Kaitlin M. Dailey, Amanda E. Brooks.

**Data curation:** Kaitlin M. Dailey, Jessica E. Pullan, Krysten E. Vance, Amanda E. Brooks.

**Formal analysis:** Kaitlin M. Dailey, James M. Small, Jessica E. Pullan, Seth Winfree, Megan Orr, Amanda E. Brooks.

**Funding acquisition:** Kaitlin M. Dailey, Sanku Mallik, Kenneth W. Bayles, Michael A. Hollingsworth, Amanda E. Brooks.

**Investigation:** Kaitlin M. Dailey, Amanda E. Brooks.

**Methodology:** Kaitlin M. Dailey, Jessica E. Pullan, Megan Orr, Amanda E. Brooks.

**Project administration:** Kaitlin M. Dailey, Sanku Mallik, Kenneth W. Bayles, Michael A. Hollingsworth, Amanda E. Brooks.

**Resources:** Kaitlin M. Dailey, Sanku Mallik, Kenneth W. Bayles, Michael A. Hollingsworth, Amanda E. Brooks.

**Software:** Kaitlin M. Dailey, Michael A. Hollingsworth, Amanda E. Brooks.

**Supervision:** Kaitlin M. Dailey, Sanku Mallik, Kenneth W. Bayles, Michael A. Hollingsworth, Amanda E. Brooks.

**Validation:** Kaitlin M. Dailey, Amanda E. Brooks.

**Visualization:** Kaitlin M. Dailey, Seth Winfree, Amanda E. Brooks.

**Writing – original draft:** Kaitlin M. Dailey, James M. Small, Jessica E. Pullan, Amanda E. Brooks.

**Writing – review & editing:** Kaitlin M. Dailey, James M. Small, Jessica E. Pullan, Seth Winfree, Krysten E. Vance, Megan Orr, Sanku Mallik, Kenneth W. Bayles, Michael A. Hollingsworth, Amanda E. Brooks.

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
