## [Decision Letter · Decision Letter 0]

5 May 2023

PONE-D-23-07958An Intravenous Pancreatic Cancer Therapeutic: Characterization of CRISPR/Cas9n-modified Clostridium novyi-Non ToxicPLOS ONE

Dear Dr. Dailey,

Thank you for submitting your manuscript to PLOS ONE. After careful consideration, we feel that it has merit but does not fully meet PLOS ONE’s publication criteria as it currently stands. Therefore, we invite you to submit a revised version of the manuscript that addresses the points raised during the review process.

We look forward to receiving your revised manuscript.

Kind regards,

Kadiam Venkata Subbaiah, Ph.D

Academic Editor

PLOS ONE

Journal Requirements:

"We acknowledge funding support for the research within this publication from the Center for Diagnostic and Therapeutic Strategies in Pancreatic Cancer at North Dakota State University, as well as discretionary funds from MAH and KWB at UNMC. Further funding was provided by a Nebraska Department of Health and Human Services LB506 to MAH, KWB, and KMD. We also gratefully acknowledge the Geographical Management of Cancer Health Disparities Program, Region 3 hosted by the University of New Mexico Comprehensive Cancer Center for covering costs associated with publication.

 Special thanks to the NDSU Agriculture Department’s Advanced Imaging Microscopy Laboratory, specifically Dr. Pawel Borowitz and Jordan Flaten, as well as the Electron Microscopy Core for TEM studies, and the Research Operations Recharge Center and Characterization Service Center for XPS analysis. Further thanks to the UNMC Tissue Sciences Facility as well as the UND Histology Core for their services in preparing the histology slides within this publication. "

"We acknowledge funding support for the research within this publication from the Center for Diagnostic and Therapeutic Strategies in Pancreatic Cancer at North Dakota State University 1P20GM109024 to SM and AEB, as well as discretionary funds from MAH and KWB at UNMC. Further funding was provided by a Nebraska Department of Health and Human Services LB506 to MAH, KWB, and KMD. We also gratefully acknowledge the Geographical Management of Cancer Health Disparities Program, Region 3 hosted by the University of New Mexico Comprehensive Cancer Center for covering costs associated with publication.

5. Please amend your list of authors on the manuscript to ensure that each author is linked to an affiliation. Authors’ affiliations should reflect the institution where the work was done (if authors moved subsequently, you can also list the new affiliation stating “current affiliation:….” as necessary).

8. PLOS ONE now requires that authors provide the original uncropped and unadjusted images underlying all blot or gel results reported in a submission’s figures or Supporting Information files. This policy and the journal’s other requirements for blot/gel reporting and figure preparation are described in detail at https://journals.plos.org/plosone/s/figures#loc-blot-and-gel-reporting-requirements and https://journals.plos.org/plosone/s/figures#loc-preparing-figures-from-image-files. When you submit your revised manuscript, please ensure that your figures adhere fully to these guidelines and provide the original underlying images for all blot or gel data reported in your submission. See the following link for instructions on providing the original image data: https://journals.plos.org/plosone/s/figures#loc-original-images-for-blots-and-gels. 

Additional Editor Comments:

RE: PONE-D-23-07958 "An Intravenous Pancreatic Cancer Therapeutic: Characterization of CRISPR/Cas9n-modified Clostridium novyi-Non Toxic"

Dear Dr. Kaitlin M Dailey,

I am pleased to inform you that the above-referenced manuscript is of interest and potentially acceptable for publication, but a major revision is required to meet the concerns of the reviewers. The reviewers' comments are provided below.

Reviewer-1

This report describes an original piece of work which could have clinical application. No one before engineering C.Novyi to express cargos. The idea of enhancing localization is very good.

My only reservations are the amount of detail (or rather lack of it) provided for the engineering. The authors need to provide more details of how they achieved their tasks and provide undisputed evidence that the constructs function as expected. I would encourage authors to expand the method section.

The use of Crispr/cas9 in Clostridium is not original itself as there is around 50 papers in the area already, but C.Novyi engineering is definitely poorly explored.

1. I find the genetic engineering part a bit sparsely written - there should be an image that shows the construction of the CRISPR/Cas plasmid, and analysis of gel electrophoresis on the five colonies (the digested PCR product) - I would also refer to Sanger sequencing to confirm that this was truly the RGD peptide sequence with no mutations. Also a control PCR should be perform to establish CRISPR/Cas plasmid loss - to make sure that the amplified PCR product does not originate from a plasmid template, but from the chromosome. Some genomic extractions also pull plasmids accidentally.

2. As I understand, expression of RGD on the spore coat is intended to help with the localization of the spores at the tumor site - few more sentences can be added in the introduction section to make it clear why this strategy was chosen, how it works, and why the study was conducted.

3. For the investigation of tissues which contained spores - a heat treatment should be performed to truly account for spores and eliminate potential counts of spores that might have germinated during the animal experiment.

Reviewer-2:

In this manuscript the authors investigate targeting Clostridium novyi bacterium to the tumor by expressing RGD peptide. This is interesting work and has the potential to provide further impetus to the area of bacterial therapeutics. However, there are significant concerns regarding some of the data and the manner of representation that needs to be addressed:

1. Line 121: The authors used PCR amplification and digestion with restriction endonucleases to confirm the genomic insertion of RGD peptide. However, performing Sanger sequencing on the PCR product would allow for a more accurate confirmation that the intended gene has been inserted without any variation.

2. Line 127, 132: Supplementary tables are not used chronologically in the manuscript.

3. Line 137: In the results section, authors mention that confocal images were taken at 40x while in methods, they mention the images were taken at 4x. The authors did not clarify how many images were taken. Quantitating multiple fields of view and depths (z-axis) of the same slide in addition to multiple replicates is critical to get a reliable result.

4. Figure 1: Can the authors clarify in the results and the figure, the comparison group for Candidate A against which significance was checked?

5. Line 191-200: The authors state an increase in pancreas percent mass (denoting pancreatic inflammation and successful spore localization) in mice injected with unmodified spores, RGD-modified spores, and PBS. However, the authors do not mention if these observations are statistically significant. If not, then drawing major inferences from this data would be incorrect.

6. Figures S6-12: The authors did not use a positive control in the gels.

7. Line 207-223: In the RGD-modified groups, why wasn’t PCR performed with primers specific to RGD? With potential cross contamination issues observed, using RGD-specific primers would confirm that the treatment group was indeed treated as well as the absence of RGD-modified C. novyi-NT in the control groups.

8. Figure 4: It is unclear if the results are statistically significant.

9. Figure 4C-D: Why is the RGD-modified C. novyi burden in the pancreas of sham mice (Fig 4C) objectively higher (~40 vs. 30) than pancreas of KPC-implanted mice (Fig 4D)? This indicates non-specificity of the therapeutic to the cancer.

10. Figure 4C-D: Can the authors combine the figure 4C and 4D so comparisons can be made about the specificity of the probe.

11. The authors suggest increase in bioburden at 24h. Comparisons between bioburden shortly after injection vs. 24h would be important.

Reviewers' comments:

Reviewer's Responses to Questions

**Comments to the Author**

1. Is the manuscript technically sound, and do the data support the conclusions?

Reviewer #1: No

Reviewer #2: Partly

2. Has the statistical analysis been performed appropriately and rigorously? 

Reviewer #1: No

Reviewer #2: N/A

3. Have the authors made all data underlying the findings in their manuscript fully available?

Reviewer #1: Yes

Reviewer #2: No

4. Is the manuscript presented in an intelligible fashion and written in standard English?

Reviewer #1: Yes

Reviewer #2: Yes

5. Review Comments to the Author

Reviewer #1: In this manuscript the authors investigate targeting Clostridium novyi bacterium to the tumor by expressing RGD peptide. This is interesting work and has the potential to provide further impetus to the area of bacterial therapeutics. However, there are significant concerns regarding some of the data and the manner of representation that needs to be addressed:

1. Line 121: The authors used PCR amplification and digestion with restriction endonucleases to confirm the genomic insertion of RGD peptide. However, performing Sanger sequencing on the PCR product would allow for a more accurate confirmation that the intended gene has been inserted without any variation.

2. Line 127, 132: Supplementary tables are not used chronologically in the manuscript.

3. Line 137: In the results section, authors mention that confocal images were taken at 40x while in methods, they mention the images were taken at 4x. The authors did not clarify how many images were taken. Quantitating multiple fields of view and depths (z-axis) of the same slide in addition to multiple replicates is critical to get a reliable result.

4. Figure 1: Can the authors clarify in the results and the figure, the comparison group for Candidate A against which significance was checked?

5. Line 191-200: The authors state an increase in pancreas percent mass (denoting pancreatic inflammation and successful spore localization) in mice injected with unmodified spores, RGD-modified spores, and PBS. However, the authors do not mention if these observations are statistically significant. If not, then drawing major inferences from this data would be incorrect.

6. Figures S6-12: The authors did not use a positive control in the gels.

7. Line 207-223: In the RGD-modified groups, why wasn’t PCR performed with primers specific to RGD? With potential cross contamination issues observed, using RGD-specific primers would confirm that the treatment group was indeed treated as well as the absence of RGD-modified C. novyi-NT in the control groups.

8. Figure 4: It is unclear if the results are statistically significant.

9. Figure 4C-D: Why is the RGD-modified C. novyi burden in the pancreas of sham mice (Fig 4C) objectively higher (~40 vs. 30) than pancreas of KPC-implanted mice (Fig 4D)? This indicates non-specificity of the therapeutic to the cancer.

10. Figure 4C-D: Can the authors combine the figure 4C and 4D so comparisons can be made about the specificity of the probe.

11. The authors suggest increase in bioburden at 24h. Comparisons between bioburden shortly after injection vs. 24h would be important.

Reviewer #2: This report describes an original piece of work which could have clinical application. No one before engineering C.Novyi to express cargos. The idea of enhancing localization is very good.

My only reservations are the amount of detail (or rather lack of it) provided for the engineering. The authors need to provide more details of how they achieved their tasks and provide undisputed evidence that the constructs function as expected. I would encourage authors to expand the method section.

The use of Crispr/cas9 in Clostridium is not original itself as there is around 50 papers in the area already, but C.Novyi engineering is definitely poorly explored.

1. I find the genetic engineering part a bit sparsely written - there should be an image that shows the construction of the CRISPR/Cas plasmid, and analysis of gel electrophoresis on the five colonies (the digested PCR product) - I would also refer to Sanger sequencing to confirm that this was truly the RGD peptide sequence with no mutations. Also a control PCR should be perform to establish CRISPR/Cas plasmid loss - to make sure that the amplified PCR product does not originate from a plasmid template, but from the chromosome. Some genomic extractions also pull plasmids accidentally.

2. As I understand, expression of RGD on the spore coat is intended to help with the localization of the spores at the tumor site - few more sentences can be added in the introduction section to make it clear why this strategy was chosen, how it works, and why the study was conducted.

3. For the investigation of tissues which contained spores - a heat treatment should be performed to truly account for spores and eliminate potential counts of spores that might have germinated during the animal experiment.

6. PLOS authors have the option to publish the peer review history of their article (what does this mean?). If published, this will include your full peer review and any attached files.

Reviewer #1: No

Reviewer #2: No

---

## [Author Response · Author response to Decision Letter 0]

21 Jun 2023

Author’s Responses to Reviewer Comments (bolded text)

We are greatly encouraged that both reviewers see the potential of our work as reported in this manuscript has to impact the development of C. novyi-NT oncotherapeutics, bacterial-mediated therapeutics, and the field of biologic therapeutics in general. We thank the reviewers for the time they clearly spent carefully critiquing of this manuscript to make it a stronger overall study. Each comment has been carefully considered by our team with revisions made where applicable. We would like to directly address the following comments:

Review Comments to the Author

Reviewer #1: In this manuscript the authors investigate targeting Clostridium novyi bacterium to the tumor by expressing RGD peptide. This is interesting work and has the potential to provide further impetus to the area of bacterial therapeutics. However, there are significant concerns regarding some of the data and the manner of representation that needs to be addressed:

1. Line 121: The authors used PCR amplification and digestion with restriction endonucleases to confirm the genomic insertion of RGD peptide. However, performing Sanger sequencing on the PCR product would allow for a more accurate confirmation that the intended gene has been inserted without any variation.

We agree with the reviewer that traditionally, Sanger Sequencing would allow for more accurate confirmation, however the GC rich nature (>60%) of this gene in addition to the poor characterization of the genome and access to primers poses a significant complication leading to better validation through our alternative implementation of careful restriction digest schemes in the methodology. Details of our methodology have been peer reviewed and published here: https://doi.org/10.3389/fmicb.2021.624618

2. Line 127, 132: Supplementary tables are not used chronologically in the manuscript.

We thank the reviewer for catching our mistake and have corrected the Supplementary Tables/Figures to match the chronological order in the manuscript. Supplementary Tables/Figures are now listed as they appear chronologically in the manuscript starting in the Methods Section.

3. Line 137: In the results section, authors mention that confocal images were taken at 40x while in methods, they mention the images were taken at 4x. The authors did not clarify how many images were taken. Quantitating multiple fields of view and depths (z-axis) of the same slide in addition to multiple replicates is critical to get a reliable result.

We thank the reviewer for catching this typo and have corrected the Methods section. We have also further clarified that the entire integrin-coated surface was quantified: “…40X scanning brightfield confocal microscopy images of the entire integrin-coated surface were obtained after crystal violet staining”. The resulting quantitative data is reported as the mean with standard deviation of 6 integrin coated surfaces per treatment repeated three iterations for a sum total n of 18. We have added this information to the Method text in addition to where it original appeared in the Figure 1 Legend. Although traditional confocal provides a z-axis, in the case of brightfield imaging as performed by the core facility did not give rise to any additional quantifiable data.

4. Figure 1: Can the authors clarify in the results and the figure, the comparison group for Candidate A against which significance was checked?

We thank the reviewer for indicating we did not clearly communicate our findings in Figure 1. We have made the following changes to the Method and the Figure 1 Legend text: “Candidate A demonstrates a higher rate of adhesion by the presence of more spores remaining on the surface when compared to any other cohort, including WT, NT and modification Candidate B.”

5. Line 191-200: The authors state an increase in pancreas percent mass (denoting pancreatic inflammation and successful spore localization) in mice injected with unmodified spores, RGD-modified spores, and PBS. However, the authors do not mention if these observations are statistically significant. If not, then drawing major inferences from this data would be incorrect.

The population of the cohorts employed in this study are included in Figure 3B, we have edited the Figure 3 Legend for clarity as follows: “The average percent weight of tumors harvested from murine cohorts in 3B 24hrs after tail vein injections”. The manuscript outlines a rigorous basic science study with capstone preclinical testing in which statistical significance cannot be claimed from the preclinical tests because of limited cohort sizes, however we believe this study is still has important implications for further surface modifications of C. novyi-NT and oncolytic bacterial in general towards the ultimate goal of clinical translation. The appropriate statistical testing was performed on all experiments where possible and validated by our biostatistician, Dr. Megan Orr, an author on this manuscript.

6. Figures S6-12: The authors did not use a positive control in the gels.

The novelty of this approach precludes the use of a characterized positive controls for the cloning; however, each of the primer sets as well as the unmodified C. novyi-NT were thoroughly evaluated. Amplicons were observed to be the appropriate size as previously published and the presence of at least one amplicon on each gel indicates the functionality of all reagents as well as equipment.

7. Line 207-223: In the RGD-modified groups, why wasn’t PCR performed with primers specific to RGD? With potential cross contamination issues observed, using RGD-specific primers would confirm that the treatment group was indeed treated as well as the absence of RGD-modified C. novyi-NT in the control groups.

We agree that PCR with further targeted primers specific to the genomically modified section would strengthen the conclusions of this manuscript. However, the genomic RGD-modification is very short as it is only six amino acids. Primers targeting this region were heavily GC rich, creating hairpins with a melting temperature exceeding the denaturation temperature, leading to our alternative strategy of PCR amplification with restriction digest as both a specific and accepted methodology for clinical detection of C. novyi. Compounding this complication, we administered an intentionally low initial dose of 1x105 spores intravenously (5000x lower than previous reports). Part of the purpose of this study was to determine if intravenous delivery was possible at a much lower dosage than had previously been reported regardless of surface modification, and to probe how spore surface modification could impact tumor localization/accumulation. Further details regarding our confirmational practices regarding genomic modification have been cited to our peer reviewed publication: https://doi.org/10.3389/fmicb.2021.624618.

8. Figure 4: It is unclear if the results are statistically significant.

The population of the cohorts employed in this study are included in Figure 3B, with the sample number stated throughout the manuscript and in all figure legends. We have updated the Figure 4 Legend to include the statement: “Cohorts employed in this study are in Figure 3B”. The manuscript outlines a rigorous basic science study with capstone preclinical testing in which statistical significance cannot be claimed from the preclinical tests because of limited cohort sizes, however we believe this study is still has important implications for further surface modifications of C. novyi-NT and oncolytic bacterial in general towards the ultimate goal of clinical translation. The appropriate statistical testing was performed on all experiments where possible and validated by our biostatistician, Dr. Megan Orr, an author on this manuscript.

9. Figure 4C-D: Why is the RGD-modified C. novyi burden in the pancreas of sham mice (Fig 4C) objectively higher (~40 vs. 30) than pancreas of KPC-implanted mice (Fig 4D)? This indicates non-specificity of the therapeutic to the cancer.

This study was not designed to test the specificity of the probe but rather the general alterations in biodistribution that C. novyi-NT spore surface modification gives rise to. We agree with the reviewer that one possible explanation of the data in Figure 4 is an alteration to the tumor-specific localization. However, as we state in the Discussion “While the presence of 16S rRNA is specific to Clostridium novyi, and thus adequate to demonstrate biodistribution of the bacterial species in general, it does not indicate whether the injected spores have (1) germinated to the vegetative phase, (2) remained as spores, or (3) have been phagocytosed.” An alternative explanation for this data could also be that the RGD modification has extended the overall circulation time of C. novyi-NT spores as has been thoroughly documented and reported in the parallel field nanoparticle literature. Testing these hypotheses, while currently underway, is beyond the scope of this submitted manuscript.

10. Figure 4C-D: Can the authors combine the figure 4C and 4D so comparisons can be made about the specificity of the probe.

This study was not designed to test the specificity of the probe but rather the general alterations in biodistribution that C. novyi-NT spore surface modification gives rise to, therefore to combine Figure 4C and D would be misrepresentation of the data causing the reader to erroneously compare the two data sets.

11. The authors suggest increase in bioburden at 24h. Comparisons between bioburden shortly after injection vs. 24h would be important.

This study probes the biodistribution of RGD-modified C. novyi-NT spores to characterize the effects of surface modification on oncolytic capacities and biodistribution. 24hrs was selected as this is the maximum amount of time circulation (and subsequently tumor localization/accumulation) can be allowed prior to the inherent replication time. We carefully formulated this study time point by leveraging our expertise in pharmaceutical sciences and incorporating our data with previous publications as cited throughout the manuscript regarding intravenous administration of C. novyi-NT to test the overall hypothesis of this study. Assessing bioburden at 1hr after inoculation would therefore be preemptive to the goal of this study and incur unjustifiable animal sacrifice. It is not clear what the clearance rate is in an immunocompetent model such as we employed in this study. If the spore is cleared prior to germination, there will be no effect on overall tumor viability. Thus, only spores that remain in the body long enough to localize to the exquisitely specific microenvironment of the tumor and germinate/replicate (>24hrs) are relevant. Further, assessing bioburden after 24hours would allow for spores to not only germinate, but to replicate as this is a live biologic therapeutic. Such replication invalidates any biodistribution calculations based on an initial dose. 

Reviewer #2: This report describes an original piece of work which could have clinical application. No one before engineering C.Novyi to express cargos. The idea of enhancing localization is very good.

My only reservations are the amount of detail (or rather lack of it) provided for the engineering. The authors need to provide more details of how they achieved their tasks and provide undisputed evidence that the constructs function as expected. I would encourage authors to expand the method section.

The use of Crispr/cas9 in Clostridium is not original itself as there is around 50 papers in the area already, but C.Novyi engineering is definitely poorly explored.

1. I find the genetic engineering part a bit sparsely written - there should be an image that shows the construction of the CRISPR/Cas plasmid, and analysis of gel electrophoresis on the five colonies (the digested PCR product) - I would also refer to Sanger sequencing to confirm that this was truly the RGD peptide sequence with no mutations. Also a control PCR should be perform to establish CRISPR/Cas plasmid loss - to make sure that the amplified PCR product does not originate from a plasmid template, but from the chromosome. Some genomic extractions also pull plasmids accidentally.

Thorough details regarding the methodology we employed to accomplish genetic engineering have been peer reviewed and published here: https://doi.org/10.3389/fmicb.2021.624618 and thus cannot by PLOS One policy be included in this manuscript. However, we have cited this publication throughout the manuscript where relevant. In reference to comments regarding Sanger Sequencing, we refer the reviewer back to our responses to Review 1 (Comments 1 and 7). In our previously published study, control experimentation was performed to establish CRISPR/Cas plasmid loss and such loss is routinely monitored. Prokaryotes, and especially Clostridial species are well known to have very low plasmid retention rates given the fundamental genomic differences from eukaryotes.

2. As I understand, expression of RGD on the spore coat is intended to help with the localization of the spores at the tumor site - few more sentences can be added in the introduction section to make it clear why this strategy was chosen, how it works, and why the study was conducted.

We thank the reviewer for pointing out an area in which we did not communicate as clearly as we intended. We have revisited the Introduction to add the following statements and references: “In this study, CRISPR/Cas9 was used to modify the genome of C. novyi to elicit the insertion, expression, and spore coat insertion of the tumor targeting tri-peptide, Arg-Gly-Asp (RGD). In previous studies in the field of nanoparticle drug delivery, tumor accumulation was improved through the integration of the RGD motif(25,26). RGD is has a well-characterized affinity for the αvβ3 integrin commonly overexpressed on tumor cells and tumor associated epithelium(26,27) including pancreatic tumors(28). Further, transient exogenous plasmid expression of RGD on the surface of other oncolytic bacterial species has altered biodistribution advantageously(7,13,29). We have previously accomplished genomic editing to insert sequence of known tumor targeting tag ‘RGD’ into C. novyi-NT and confirmed retained oncolytic capacities(20). This manuscript describes the incorporation of RGD into the spore coat, including probing biodistribution and immune stimulation of modified C. novyi-NT in a murine pancreatic tumor model with an intact immune system.”

3. For the investigation of tissues which contained spores - a heat treatment should be performed to truly account for spores and eliminate potential counts of spores that might have germinated during the animal experiment.

Given that the vegetative form of C. novyi-NT cannot survive in oxygenated levels above approximately 0.3% p02 (which is exclusively found physiologically in the tumor microenvironment), heat treatment is not necessary as the hypoxic environment critical to sustaining vegetative cells is destroyed during the necropsy process. Heat treatment does not affect the presence or absence of spores and may in fact cause them to germinate when they should not, providing erroneous conclusions.

Other comments to the Authors:

1. Is the manuscript technically sound, and do the data support the conclusions?

Reviewer #1: No

Reviewer #2: Partly

We thank the reviewers for their critiques and have addressed all comments regarding scientific rigor and technical soundness of the data presented as raised by reviewers above.

2. Has the statistical analysis been performed appropriately and rigorously?

Reviewer #1: No

Reviewer #2: N/A

The manuscript outlines a rigorous basic science study with capstone preclinical testing in which statistical significance cannot be claimed from the preclinical tests because of limited cohort sizes, however we believe this study is still has important implications for further surface modifications of C. novyi-NT and oncolytic bacterial in general towards the ultimate goal of clinical translation. The appropriate statistical testing was performed on all experiments where possible and validated by our biostatistician, Dr. Megan Orr, an author on this manuscript.

3. Have the authors made all data underlying the findings in their manuscript fully available?

The PLOS Data policy [plosone.org] requires authors to make all data underlying the findings described in their manuscript fully available without restriction, with rare exception (please refer to the Data Availability Statement in the manuscript PDF file). The data should be provided as part of the manuscript or its supporting information, or deposited to a public repository. For example, in addition to summary statistics, the data points behind means, medians and variance measures should be available. If there are restrictions on publicly sharing data—e.g. participant privacy or use of data from a third party—those must be specified.

Reviewer #1: Yes

Reviewer #2: No

The histology datasets generated during and/or analyzed during the current study will be made available upon acceptance for publication in the repository: The Cancer Imaging Archive hosted by the National Cancer Institute, https://www.cancerimagingarchive.net/. Further datasets generated during and/or analyzed during the current study are included in the supporting information.

---

## [Decision Letter · Decision Letter 1]

13 Jul 2023

An Intravenous Pancreatic Cancer Therapeutic: Characterization of CRISPR/Cas9n-modified Clostridium novyi-Non Toxic

PONE-D-23-07958R1

Dear Dr. Kaitlin,  

We’re pleased to inform you that your manuscript has been judged scientifically suitable for publication and will be formally accepted for publication once it meets all outstanding technical requirements.

Kind regards,

Kadiam C Venkata Subbaiah, Ph.D

Academic Editor

PLOS ONE

Additional Editor Comments (optional):

Reviewers' comments:

Reviewer's Responses to Questions

**Comments to the Author**

1. If the authors have adequately addressed your comments raised in a previous round of review and you feel that this manuscript is now acceptable for publication, you may indicate that here to bypass the “Comments to the Author” section, enter your conflict of interest statement in the “Confidential to Editor” section, and submit your "Accept" recommendation.

Reviewer #1: All comments have been addressed

Reviewer #3: All comments have been addressed

2. Is the manuscript technically sound, and do the data support the conclusions?

Reviewer #1: Yes

Reviewer #3: Yes

3. Has the statistical analysis been performed appropriately and rigorously? 

Reviewer #1: N/A

Reviewer #3: Yes

4. Have the authors made all data underlying the findings in their manuscript fully available?

Reviewer #1: Yes

Reviewer #3: Yes

5. Is the manuscript presented in an intelligible fashion and written in standard English?

Reviewer #1: Yes

Reviewer #3: Yes

6. Review Comments to the Author

Reviewer #1: (No Response)

Reviewer #3: The authors have addressed reviewers comments/concerns and improved the quality of manuscript. However, there are some minor errors that needed to be correct before I recommend publication of this manuscript.

1) Wording and spacing inconsistencies throughout the revised manuscript, especially the words between

• “ hours ” and “ hrs ” with spacing or no spacing, including a word “min”

• “ 16s RNA ” and “ 16S RNA ”

2) In revised manuscript with track changes

• Line 21-22 : changes “ ; ” to “ , ” and delete the comma before Brooks,2

• Line 24-27 : delete “ ; ” after “NE” and “ND”

• Line 45 : no spacing between “ necrotic-impeding ”

• Line 162 : add “ nm ” after sentence “ a wavelength of 590 ” and formatting front to Italic in line 162 and 166 “ C. novyi ”

• Line 231 and 242 : add “ and ” in (Fig 4A and Supporting Information)

• Line 457 : remove Italic format “ twice ”

• Line 500-514 : add “ and ” in (Fig 5E and 5F)

• Line 524-525 : changes wording for Fig X to Figure X

• Line 666 : changes equation (1) font format to Not Bold

• Line 657 and 710 : add spacing “ at 150 rpm ” and “ 3-5 days ”

• Line 739 and 753 : add spacing “ at 12,000 rpm ” and “ to 19 ng ”

• Line 808 : changes “ Fig 2B-C ” to “ Fig 2B and 2C ”

• Line 809-810 : changes wording for Fig X to Figure X

3) In Supporting information captions

• Line 017 : deletes “ s ” in a word “ Figure S12 ”

7. PLOS authors have the option to publish the peer review history of their article (what does this mean?). If published, this will include your full peer review and any attached files.

Reviewer #1: No

Reviewer #3: No

---

## [Editor Report · Acceptance letter]

19 Jul 2023

PONE-D-23-07958R1 

An Intravenous Pancreatic Cancer Therapeutic: Characterization of CRISPR/Cas9n-modified *Clostridium novyi*-Non Toxic 

Dear Dr. Dailey:

I'm pleased to inform you that your manuscript has been deemed suitable for publication in PLOS ONE. Congratulations! Your manuscript is now with our production department. 

Kind regards, 

on behalf of

Dr. Kadiam C Venkata Subbaiah 

Academic Editor

PLOS ONE